# Stochastic activity in low-rank recurrent neural networks

**Francesca Mastrogiuseppe**[ID][1]*, **Joana Carmona**[1], **Christian K. Machens**[1]

Champalimaud Foundation, Neuroscience Research Programme, Lisbon, Portugal

* francesca.mastrogiuseppe@research.fchampalimaud.org

**Data availability statement:** The code for reproducing the figures is available at https://github.com/fmastrogiuseppe/StochasticLowRank.

## Abstract

The geometrical and statistical properties of brain activity depend on the way neurons connect to form recurrent circuits. However, the link between connectivity structure and emergent activity remains incompletely understood. We investigate this relationship in recurrent neural networks with additive stochastic inputs. We assume that the synaptic connectivity can be expressed in a low-rank form, parameterized by a handful of connectivity vectors, and examine how the geometry of emergent activity relates to these vectors. Our findings reveal that this relationship critically depends on the dimensionality of the external stochastic inputs. When inputs are low-dimensional, activity remains low-dimensional, and recurrent dynamics influence it within a subspace spanned by a subset of the connectivity vectors, with dimensionality equal to the rank of the connectivity matrix. In contrast, when inputs are high-dimensional, activity also becomes potentially high-dimensional. The contribution of recurrent dynamics is apparent within a subspace spanned by the totality of the connectivity vectors, with dimensionality equal to twice the rank of the connectivity matrix. Applying our formalism to excitatory-inhibitory networks, we discuss how the input configuration also plays a crucial role in determining the amount of amplification generated by non-normal dynamics. Our work provides a foundation for studying activity in structured brain circuits under realistic noise conditions, and offers a framework for interpreting stochastic models inferred from experimental data.

## Author Summary

Low-rank recurrent networks have recently emerged as a mathematically tractable framework for studying the relationship between connectivity and activity in biological and artificial neural circuits. Those models naturally produce low-dimensional activity patterns, consistent with brain recordings during cognitive tasks. However, they have so far been studied in settings with highly simplified external inputs, limiting their applicability to biologically relevant scenarios. In this work, we investigate the dynamics of low-rank networks driven by noisy inputs with varying geometries. We find that when inputs are high-dimensional, the statistics and geometry of the

**Funding:** FM is supported by a Transition To Independence grant from the Simons Foundation Collaboration on the Global Brain (code 00002313). JC is supported by a PhD studentship from Fundação para a Ciência e a Tecnologia (DOI: 10.54499/2020.05021.BD). CKM is supported by the Simons Foundation Collaboration on the Global Brain (code 543009 and 2794-04), as well as NIH R01 EY035896 and NIH RF1 NS127107. The funders had no role in study design, data collection and analysis, decision to publish, or preparation of the manuscript.

**Competing interests:** The authors have declared that no competing interests exist.

resulting activity differ markedly from previous descriptions. In particular, activity can become high-dimensional. Among the many dimensions it spans, those shaped by recurrent interactions are both more numerous and structurally distinct compared to those in networks receiving simpler inputs. While some of these directions encode input amplification by recurrent connectivity, others reflect input suppression. By extending the low-rank framework to more realistic settings, our work opens new avenues for applications in data analysis, and in modeling learning and variability in cortical circuits.

## Introduction

Understanding the relationship between connectivity and activity in neural circuits is a central focus of theoretical neuroscience. A particularly active area of research investigates how large-scale circuits can give rise to low-dimensional activity patterns [1–3]. This type of activity is commonly observed in neural recordings during behavioral tasks, particularly in cortical areas involved in high-level functions like motor planning and decision-making [4–6].

Recurrent neural networks (RNNs) with low-rank connectivity have emerged as a flexible and mathematically-tractable framework for studying this phenomenon [7–10]. Low-rank matrices are characterized by a simple structure: they are defined by a finite number of non-zero singular values and can be constructed from a small set of connectivity vectors. This simplicity facilitates the study of the relationship between connectivity and activity, offering insights that are harder to extract from generic models [1,11–14].

Previous studies on low-rank RNNs have primarily focused on models with autonomous dynamics or specific types of external inputs. These inputs were assumed to have a simple and smooth temporal structure (e.g. constant [8] or sinusoidal [15]). Furthermore, they were low-dimensional: the inputs received by different neurons were highly correlated over time, as they originated from a small set of temporal signals broadcast across the entire network. Under these conditions, low-rank RNNs have been shown to generate low-dimensional activity, whose statistical properties can be directly traced back to the structure of the network's external inputs and recurrent connectivity [8,13].

Far less is known about the behavior of low-rank RNNs in response to rapidly varying and stochastic inputs, which are routinely used to model trial-to-trial variability in circuit models [16–18]. These inputs can be either low-dimensional, when they are generated from a small set of stochastic signals and exhibit high correlations across units, or high-dimensional, when they arise from a large number of stochastic signals and remain uncorrelated across units.

Investigating the dynamics of low-rank RNNs in response to this type of input is important for several reasons. First, stochastic inputs naturally arise in cortical circuits from irregular spiking activity and the stochastic nature of biological processes like synaptic transmission [19,20]. Understanding how these inputs influence the emergence of low-dimensional activity in recurrent circuits is therefore an important biological question. Second, stochastic inputs are routinely used during the optimization of RNNs on behavioural tasks [21,22], where they help regularize the learning dynamics [23,24] and improve the robustness of internal representations [25]. As optimization often results in approximately low-rank connectivity [26,27], developing tools for describing stochastic activity in low-rank RNNs is essential for studying learning. Finally, statistical models for inferring circuit dynamics from recorded neural activity often explicitly incorporate stochastic terms [28–30]. Understanding activity in stochastic models is therefore essential for interpreting the results of fitting procedures.

In this work, we study the emergent activity in linear, low-rank recurrent neural networks driven by fast stochastic inputs, exploring both low- and high-dimensional input regimes.

We analyze the statistical and geometrical properties of activity by examining its covariance matrix, which is expressed in terms of the vectors representing both the external inputs and the low-rank connectivity. Crucially, we find that the functional dependence between the covariance matrix and these vectors undergoes a qualitative shift depending on the dimensionality of the external stochastic inputs. For low-dimensional inputs, activity covariance is also low-dimensional, and depends on the inputs together with a subset of the connectivity vectors, whose dimensionality matches the rank of the connectivity matrix. In this regime, activity qualitatively resembles that previously described in deterministic models [8,13]. In contrast, for high-dimensional inputs, activity covariance is potentially high-dimensional. The covariance depends on the inputs, which are high-dimensional, and on the full set of connectivity vectors, whose dimensionality equals twice the rank of the connectivity matrix. Part of the connectivity vectors signal directions along which inputs are amplified, and part directions along which inputs are suppressed. This regime qualitatively differs from previously reported ones, as the full geometrical structure of synaptic connectivity is reflected in the activity covariance. Our findings provide insights into how structured connectivity shapes neural activity in the presence of noise, laying the groundwork for a deeper understanding of brain circuit dynamics.

## Results

### 1. Setup

We consider a stochastic recurrent neural network of $N$ units as in Fig 1A, governed by the linear dynamics:

$$\frac{\mathrm{d}\boldsymbol{x}(t)}{\mathrm{d}t} = -\boldsymbol{x}(t) + W\boldsymbol{x}(t) + U\boldsymbol{\chi}(t) \tag{1}$$

where, for simplicity, the activity evolution timescale is set to 1. On the right-hand side of Eq 1, the first term represents the leak, the second term models the recurrent interactions mediated by the synaptic connectivity matrix $W$, and the third term corresponds to the external stochastic input. The stochastic input is expressed in terms of a Gaussian process $\boldsymbol{\chi}(t)$, which has zero mean, is uncorrelated across units, and is temporally white. It is fed into the network through a matrix of input weights $U$. The covariance of the total stochastic input is therefore given by

$$\left\langle U\boldsymbol{\chi}(t) \left[ U\boldsymbol{\chi}(s) \right]^\top \right\rangle = \bar{\Sigma}_{\text{inp}}\delta(t - s), \tag{2}$$

where $\langle \ \cdot \ \rangle$ denotes an average over inputs realizations, and we use the notation $\bar{\Sigma}_{\text{inp}} = UU^\top$ to indicate the static part of the covariance. Our analysis focuses on the activity generated by this model in the stationary regime, which is reached provided that the real part of all eigenvalues of $W$ is less than 1, and therefore the autonomous dynamics are stable. In this regime, the activity statistics can be equivalently computed over input realizations or time.

We first consider rank-one connectivity matrices before turning to higher-rank cases. A rank-one connectivity matrix can be expressed as

$$W = k \, \boldsymbol{m}\boldsymbol{n}^\top, \tag{3}$$

where $k$ is a positive scalar and $\boldsymbol{m}$ and $\boldsymbol{n}$ are unit-norm vectors (Fig 1B). By defining the overlap function between two vectors $\boldsymbol{v}$ and $\boldsymbol{v}'$ as

$$\rho_{vv'} = \boldsymbol{v}^\top \boldsymbol{v}' \tag{4}$$

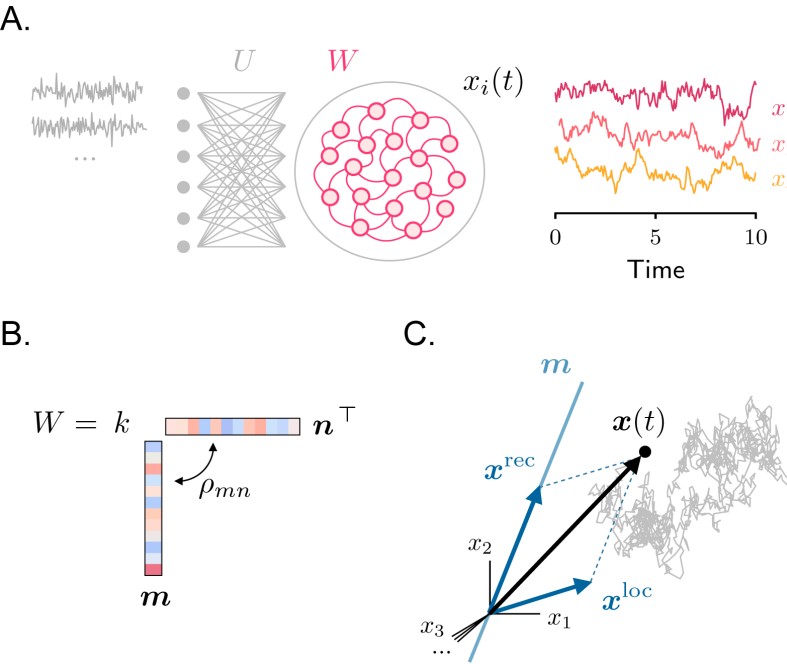

**Fig 1. Setup. A.** Left: model architecture. Right: sample activity traces from three randomly chosen neurons. **B.** Rank-one recurrent connectivity. **C.** Illustration of a sample activity trajectory in the high-dimensional space where each axis corresponds to the activity of a different neuron. Activity (black arrow) is given by the sum of two components (Eq 7, blue arrows); the direction of the component generated from recurrent interactions is fixed, and is aligned with the connectivity vector $\boldsymbol{m}$.

we therefore have $\rho_{mm} = \rho_{nn} = 1$. Vectors $\boldsymbol{m}$ and $\boldsymbol{n}$, which we refer to as *connectivity vectors*, characterize the structure of the synaptic connectivity. When $\boldsymbol{m}$ and $\boldsymbol{n}$ coincide, connectivity is symmetric. Conversely, when $\boldsymbol{m}$ and $\boldsymbol{n}$ are orthogonal, connectivity is maximally asymmetric. In general, the overlap between the two vectors is quantified by $\rho_{mn} = \boldsymbol{m}^\top \boldsymbol{n}$ (Eq 4), which ranges from –1 to 1 (symmetric matrices) passing through 0 (maximally asymmetric). The only non-zero eigenvalue of $W$ is given by $\lambda = k\rho_{mn}$, and is associated with the eigenvector $\boldsymbol{m}$.

Notably, due to normalization, the entries of the connectivity vectors scale as $O(1/\sqrt{N})$ with the network size. We further assume that the scalar $k$ is $O(1)$ in $N$, so that the eigenvalue $\lambda$ takes finite values. This scaling is consistent with the one assumed for low-rank connectivity in several previous studies [8,11,31] (see [1,10] for different choices). However, here we consider arbitrary network sizes, and do not restrict ourselves to the large-$N$ limit.

## 2. Stochastic activity in rank-one networks

We start deriving closed-form expressions for the activity generated by the dynamics in Eq 1. Integrating over time, and focusing on the stationary state, we have [32]

$$\boldsymbol{x}(t) = \int_0^t \exp[(W - I)(t - u)]U\boldsymbol{\chi}(u)\, \mathrm{d}u. \tag{5}$$

We can simplify this expression by computing the propagator operator $\exp[(W-I)t]$. For a rank-one connectivity matrix $W$, this has a simple expression (see Methods 1):

$$\exp[(W-I)t] = \exp(-t)\left[I + \frac{\exp(\lambda t) - 1}{\lambda} k\, \boldsymbol{m}\boldsymbol{n}^\top\right]. \tag{6}$$

The first term in parentheses on the right-hand side corresponds to the propagator of an unconnected network ($W = 0$), where the dynamics are driven solely by the local filtering, operated by the leak, of the external inputs. The second term captures instead the contribution to the propagator arising from the recurrent connectivity matrix $W$. By applying Eq 6 to Eq 5, we can therefore express activity as the sum of a local and a recurrent component (Fig 1C):

$$\boldsymbol{x}(t) = \boldsymbol{x}^{\mathrm{loc}}(t) + \boldsymbol{x}^{\mathrm{rec}}(t), \tag{7}$$

where we have defined

$$\boldsymbol{x}^{\mathrm{loc}}(t) = \int_0^t \exp[-(t-u)]\, U\boldsymbol{\chi}(u)\, \mathrm{d}u \tag{8}$$

$$\boldsymbol{x}^{\mathrm{rec}}(t) = k\, \boldsymbol{m}\boldsymbol{n}^\top \int_0^t \frac{\exp[-(t-u)]\{\exp[\lambda(t-u)]-1\}}{\lambda}\, U\boldsymbol{\chi}(u)\, \mathrm{d}u. \tag{9}$$

The local component, $\boldsymbol{x}^{\mathrm{loc}}$, reflects the behavior of an uncoupled RNN and represents the temporal integration of stochastic inputs performed independently by all units. Importantly, at each time point, the direction of vector $\boldsymbol{x}^{\mathrm{loc}}$ varies, depending on the direction of the most recent stochastic inputs. This activity component can therefore potentially be high-dimensional, depending on the structure of the input matrix $U$. The recurrent component, $\boldsymbol{x}^{\mathrm{rec}}$, represents instead the contribution to activity arising from the temporal integration of recurrent inputs. Unlike $\boldsymbol{x}^{\mathrm{loc}}$, the direction of $\boldsymbol{x}^{\mathrm{rec}}$ is fixed and determined by the connectivity vector $\boldsymbol{m}$.

## 2.1. Computing the activity covariance

Our goal is to characterize the statistical and geometrical properties of this emergent activity. As detailed below, these can be extracted from its covariance matrix, defined as $\Sigma \equiv \langle \boldsymbol{x}(t)\boldsymbol{x}(t)^\top\rangle$. We compute the covariance using Eqs 7, 8, 9. This calculation yields (see Methods 2):

$$\Sigma = \frac{1}{2}\left\{\bar{\Sigma}_{\mathrm{inp}} + \frac{k}{2-\lambda}\left[\bar{\Sigma}_{\mathrm{inp}}\boldsymbol{n}\boldsymbol{m}^\top + \boldsymbol{m}\boldsymbol{n}^\top\bar{\Sigma}_{\mathrm{inp}}\right] + \frac{k^2}{(2-\lambda)(1-\lambda)}\,\boldsymbol{m}\boldsymbol{n}^\top\bar{\Sigma}_{\mathrm{inp}}\,\boldsymbol{n}\boldsymbol{m}^\top\right\}. \tag{10}$$

The four additive terms on the right-hand side correspond to the covariances of the two activity components defined in Eq 7: $\boldsymbol{x}^{\mathrm{loc}}$ (first term of the covariance) and $\boldsymbol{x}^{\mathrm{rec}}$ (fourth term), as well as their cross-covariances (second and third terms). Eq 10 shows that the activity covariance is determined by matrix multiplications among the connectivity vectors and the static input covariance, indicating that the statistics of emergent activity are determined by the relative geometrical arrangement of the connectivity vectors and external inputs.

To clarify the geometrical structure of the covariance, we rewrite it as

$$\Sigma = \frac{1}{2}\left\{\bar{\Sigma}_{\mathrm{inp}} + \frac{k}{2-\lambda}\left[\boldsymbol{d}\boldsymbol{m}^\top + \boldsymbol{m}\boldsymbol{d}^\top\right] + \frac{\sigma k^2}{(2-\lambda)(1-\lambda)}\,\boldsymbol{m}\boldsymbol{m}^\top\right\} \tag{11}$$

where we have defined $d = \bar{\Sigma}_{\text{inp}}\, n$ and $\sigma = n^\top \bar{\Sigma}_{\text{inp}}\, n$. The first term on the right-hand side represents the input covariance, which can be either a low-rank or full-rank matrix, depending on whether the inputs are low- or high-dimensional. The remaining terms are rank-one matrices, spanned by the connectivity vector $m$ and the vector $d$. The direction of $d$ is jointly determined by the connectivity vector $n$ and the input covariance.

In the rest of the paper, we analyze in detail the properties of the covariance matrix in the limiting cases of one- and high-dimensional inputs. To model these two scenarios, we construct $U$ as an orthonormal matrix, but set some of its columns to zero. The number of non-vanishing columns determines the input dimensionality. We start considering the case of one-dimensional inputs, for which $U = [u, 0, \dots, 0]$, where $u$ is an $N$-dimensional input vector, and $0$ is a vector of zeros. In this case, we have that $\bar{\Sigma}_{\text{inp}} = uu^\top$, and thus, the vector $d$ is aligned with the input vector: $d = \rho_{nu} u$ (we have used Eq 4 to define the overlap between the connectivity vector $n$ and the input vector $u$, denoted as $\rho_{nu}$). We then consider the case where inputs have maximal dimensionality. In this case, $U$ has $N$ orthonormal columns and $\bar{\Sigma}_{\text{inp}}$ coincides with the identity matrix; therefore, the vector $d$ coincides with the connectivity vector $n$. In the intermediate cases, which correspond to inputs of intermediate dimensionality, the vector $d$ is given by the projection of the connectivity vector $n$ on the subspace spanned by the non-vanishing columns of $U$. This is because we can rewrite:

$$d = \sum_{c=1}^{C} \rho_{nu^c} u^c \tag{12}$$

where $\{u^c\}_c$ represent the $C$ non-vanishing columns of $U$.

We conclude this section by remarking that, in the general case, the geometrical structure of the covariance depends on both connectivity vectors, $m$ and $n$. While the dependence of activity covariance on the connectivity vector $m$ – which corresponds to the right eigenvector of the connectivity matrix – has been emphasized in previous work on low-rank RNNs [8,33,34], the dependence on the connectivity vector $n$ is perhaps more surprising. We will explore this dependence further in the subsequent sections.

## 3. One-dimensional stochastic inputs

We begin with the case of one-dimensional stochastic inputs, where a single signal is broadcast across the entire network, leading to temporally correlated inputs for individual neurons (Fig 2A). This case closely relates to previous work that examined the impact of deterministic low-dimensional inputs in low-rank RNNs [8,33].

Setting $\bar{\Sigma}_{\text{inp}} = uu^\top$ in Eq 11, the activity covariance becomes:

$$\Sigma = \frac{1}{2}\left\{ uu^\top + \frac{\rho_{nu}k}{2-\lambda}\left[ um^\top + mu^\top \right] + \frac{\rho_{nu}^2 k^2}{(2-\lambda)(1-\lambda)}\, mm^\top \right\}. \tag{13}$$

As in the general case, the first and fourth terms on the right-hand side represent the covariance of the activity components $x^{\text{loc}}$ and $x^{\text{rec}}$, while the second and third terms represent the cross-covariances between them. All terms, except the first one, are proportional to the overlap parameter $\rho_{nu}$. This implies that, when the input vector $u$ has zero overlap with the connectivity vector $n$, the network behaves as if it were unconnected. Therefore, recurrent connectivity contributes to shaping activity in the case of low-dimensional inputs only when the vectors $u$ and $n$ are characterized by a non-vanishing overlap.

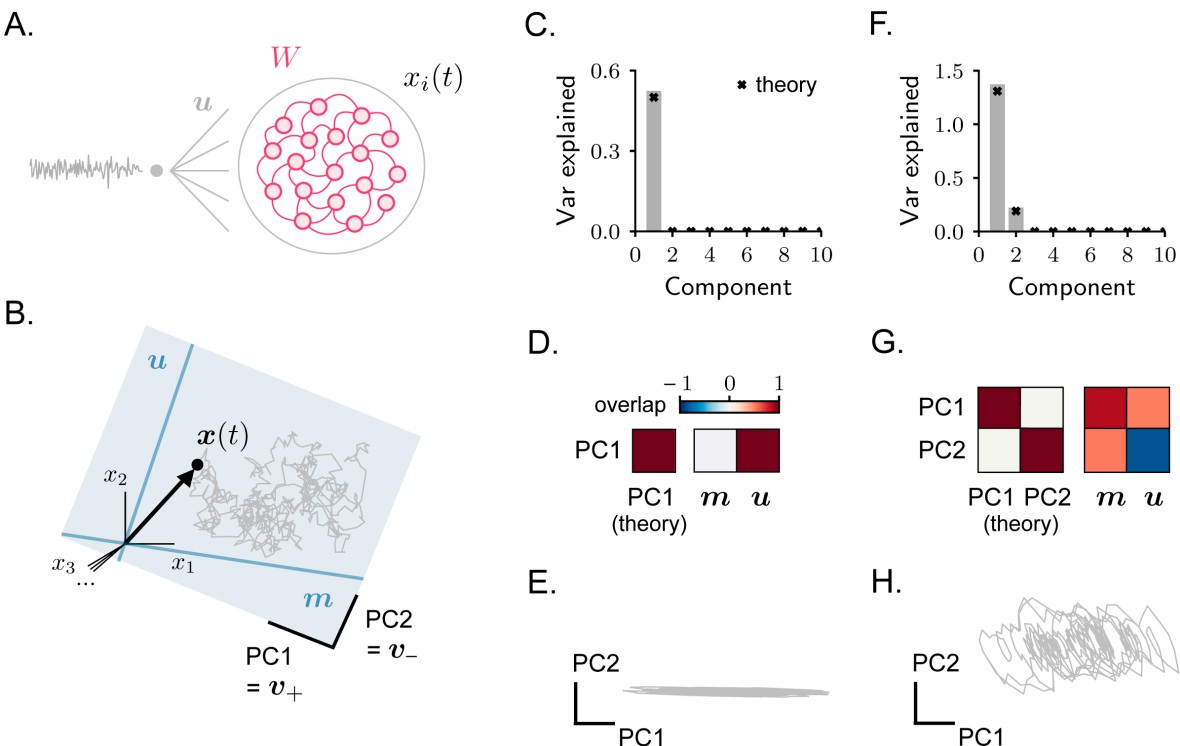

**Fig 2. Rank-one RNN receiving one-dimensional stochastic inputs. A.** Model architecture. **B.** Activity covariance is low-dimensional, and is spanned by connectivity vector $m$ together with the external input vector $u$. As a consequence, activity is contained within the plane collinear with these two vectors. **C–D–E.** Example of a simulated network with $\rho_{nu} = 0$. In C: covariance spectrum. Components larger than 10 are not displayed (they are all close to zero). In D: overlap between the dominant principal components estimated from simulated activity and the theoretically-estimated PCs (left), or the vectors $m$ and $u$ (right). Overlaps are quantified via Eq 4, with input vectors $u$ chosen to be normalized. Note that here, but not in G, only one principal component can be identified. In E: simulated activity projected on the two dominant PCs. **F–G–H.** Same as in C–D–E, example with $\rho_{nu} > 0$.

## 3.1. Covariance eigenvalues and eigenvectors

Covariance matrices provide important information on the statistical and geometrical properties of activity. One effective way to analyze these properties is by computing the covariance matrix eigenvectors and eigenvalues, which provide information on the directions and magnitude of co-variation of activity over time. The eigenvectors of the covariance matrix, denoted as $\{v_i\}_i$, are also known as the Principal Components (PCs) of activity. The corresponding eigenvalues, denoted as $\{\mu_i\}_i$, quantify the variance of activity along each principal component. From the covariance eigenvalues, the linear dimensionality of activity is commonly estimated using the participation ratio, quantifying the concentration of the eigenvalue distribution [35,36].

The covariance in Eq 13 is a rank-two matrix. Its eigenvalues can be computed through the method of reduced matrices [37] (see Methods 3), yielding:

$$\mu_{\pm} = \frac{1}{4}\left[ 1 + 2\alpha\rho_{mu}\rho_{nu} + \beta\rho_{nu}^2 \pm \sqrt{\left(1 + 2\alpha\rho_{mu}\rho_{nu} + \beta\rho_{nu}^2\right)^2 - 4\rho_{nu}^2\left(1 - \rho_{mu}^2\right)\left(\beta - \alpha^2\right)} \right] \quad (14)$$

where we have defined $\rho_{mu} = \boldsymbol{m}^\top \boldsymbol{u}$ (Eq 4), as well as the short-hand notations $\alpha = k/(2-\lambda)$ and $\beta = k^2/[(2-\lambda)(1-\lambda)]$. To find the eigenvectors associated with these eigenvalues, we can leverage the low-dimensional structure of Eq 13 to formulate the ansatz:

$$\boldsymbol{v}_\pm = \gamma_\pm \boldsymbol{m} + \boldsymbol{u}, \tag{15}$$

and compute the coefficients $\gamma_\pm$ through straightforward algebra (see Methods 3).

The fact that the covariance matrix has only two non-zero eigenvalues indicates that, in the stationary regime, the activity $\boldsymbol{x}(t)$ is confined to a two-dimensional plane, spanned by the connectivity vector $\boldsymbol{m}$ and the input vector $\boldsymbol{u}$ (Fig 2B). Within the $\boldsymbol{m}$-$\boldsymbol{u}$ plane, the eigenvectors $\boldsymbol{v}_\pm$ define the principal components of activity fluctuations. When the variance of activity fluctuations is primarily concentrated along the first PC, the covariance eigenvalue $\mu_+$ is much larger than $\mu_-$, and the activity dimensionality is close to 1; conversely, when the variance is distributed similarly along the two PCs, the two covariance eigenvalues are similar, and the dimensionality approaches 2.

We have seen that when the input vector $\boldsymbol{u}$ and the connectivity vector $\boldsymbol{n}$ are characterized by zero overlap ($\rho_{nu} = 0$), recurrent interactions do not contribute to the activity. In this case, the eigenvalues (Eq 14) read:

$$\mu_+ = \frac{1}{2}, \ \mu_- = 0. \tag{16}$$

Since the eigenvalue $\mu_-$ vanishes, activity is one-dimensional (Fig 2C and 2E). The only non-trivial PC of activity, $\boldsymbol{v}_+$, is fully aligned with the input vector $\boldsymbol{u}$, as the value of $\gamma_+$ vanishes (Fig 2D, see Methods 3). Therefore, as expected, activity reduces to a local filtering of the stochastic inputs, completely operating along the input direction.

When the input vector $\boldsymbol{u}$ and the connectivity vector $\boldsymbol{n}$ are characterized by non-zero overlap, activity is characterized by a non-vanishing component along $\boldsymbol{m}$ (Fig 2F, 2G and 2H). While the component of activity that is aligned with $\boldsymbol{u}$ reflects the integration of external inputs that is performed locally and independently by single neurons, the component aligned with vector $\boldsymbol{m}$ represents the contribution to activity from recurrent interactions. The exact orientation of the vectors $\boldsymbol{m}$ and $\boldsymbol{u}$ with respect to the two principal components depends on the overlap parameters $\rho_{mn}$, $\rho_{nu}$ and $\rho_{mu}$, specifying the alignment among the three vectors $\boldsymbol{m}$, $\boldsymbol{n}$ and $\boldsymbol{u}$.

To summarize, rank-one RNNs driven by one-dimensional stochastic inputs generate activity confined to the plane defined by the connectivity vector $\boldsymbol{m}$ and the input vector $\boldsymbol{u}$. In this setting, the contribution of recurrent connectivity to the activity covariance is therefore one-dimensional. These findings are qualitatively consistent with results obtained for non-linear rank-one RNNs responding to one-dimensional deterministic inputs [8,33]. We now turn to the opposite case of high-dimensional stochastic inputs and demonstrate that a qualitatively different behavior emerges in that regime.

## 4. High-dimensional stochastic inputs

We now consider high-dimensional stochastic inputs, for which the input to each neuron in the network is uncorrelated and has identical variance (Fig 3A, Methods 4). Using $\bar{\Sigma}_{\text{inp}} = I$ in Eq 11, the covariance becomes

$$\Sigma = \frac{1}{2}\left\{ I + \frac{k}{2-\lambda}\left[ \boldsymbol{nm}^\top + \boldsymbol{mn}^\top + \frac{k}{1-\lambda}\,\boldsymbol{mm}^\top \right] \right\}. \tag{17}$$

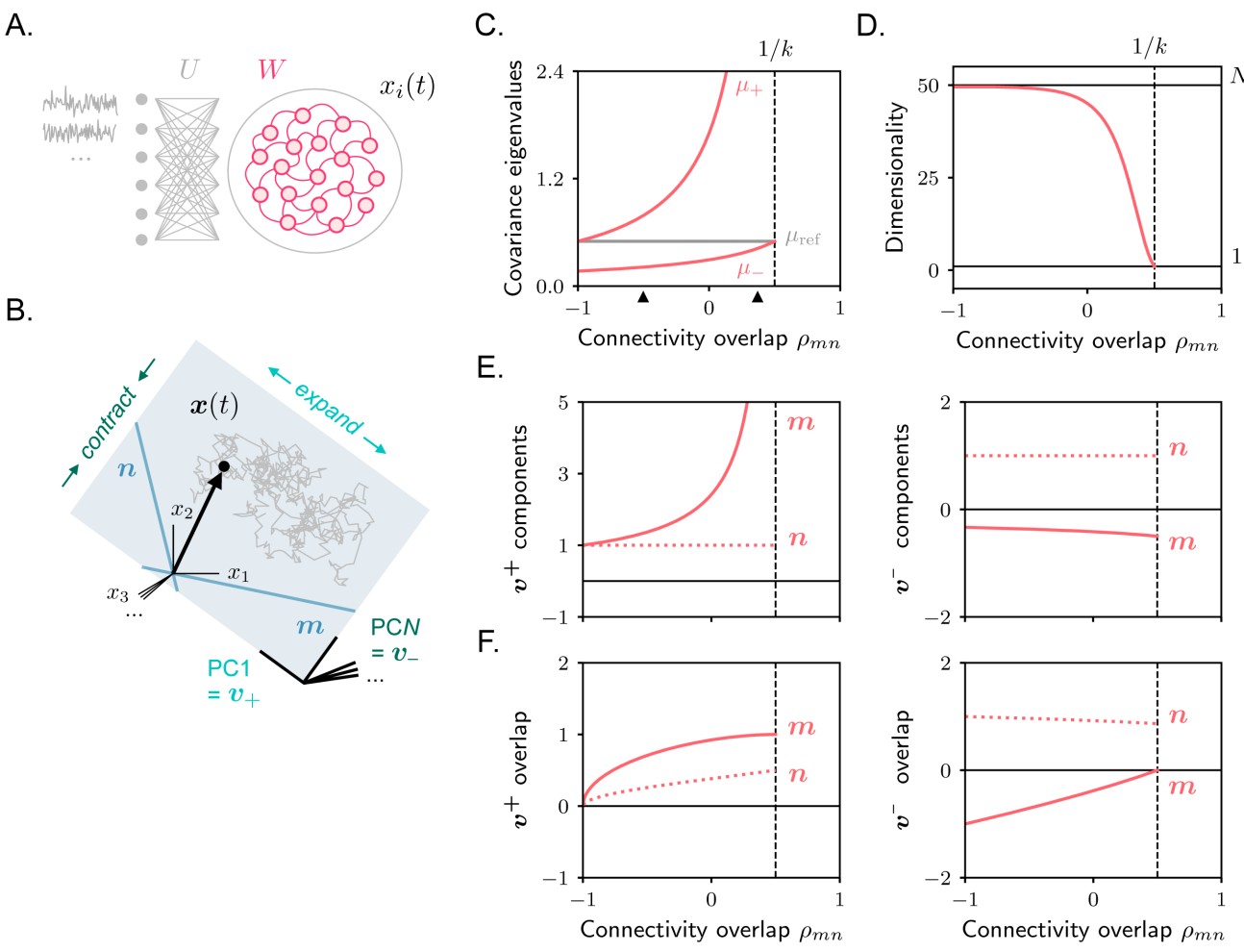

**Fig 3. Rank-one RNN receiving high-dimensional stochastic inputs. A.** Model architecture. **B.** The activity covariance is high-dimensional, with all eigenvalues taking identical values except for two – one larger and one smaller. The principal components associated with these two eigenvalues lie within the plane spanned by the connectivity vectors $m$ and $n$. **C.** Covariance eigenvalues as a function of overlap between connectivity vectors. The dashed vertical line indicates the value of $\rho_{mn}$ for which dynamics become unstable. Black arrows indicate the value of $\rho_{mn}$ that is used for simulations in Fig 4. **D.** Dimensionality. Horizontal black lines indicate the maximum ($N$) and the minimum (1) possible values. **E.** Components of $v_+$ (or PC1 vector, left) and $v_-$ (or PCN vector, right) along connectivity vectors $m$ and $n$, as from Eq 24. **F.** Overlap (Eq 4) between the principal components $v_+$ and $v_-$ (after normalization) and the connectivity vectors $m$ and $n$.

As expected for high-dimensional inputs, the covariance matrix is now full-rank. However, it exhibits a particularly simple structure: an additive low-rank perturbation, generated from the connectivity vectors $m$ and $n$, on the identity matrix. This structure enables us to derive the behavior of its eigenvalues and eigenvectors in closed form. Before delving into the derivations, we explore the covariance matrix from a different perspective.

## 4.1. Dependence on connectivity vector $n$

The covariance in Eq 17 differs from that observed in response to low-dimensional inputs in several ways, one of which is its dependence not only on the connectivity vector $m$, but also on $n$. This dependence is particularly interesting, as it contrasts with previous studies on low-rank RNNs which have primarily emphasized the dominant role of connectivity vector $m$ over $n$ in shaping the geometrical structure of activity [8,13,33]. In this section, we examine this

dependence in detail to better understand its origin. In the next section, we characterize the impact of the connectivity vector $\boldsymbol{n}$ on the covariance structure.

The covariance terms that depend on $\boldsymbol{n}$ correspond to cross-covariances among the two terms constituting activity $\boldsymbol{x}^{\text{loc}}$ and $\boldsymbol{x}^{\text{rec}}$ (Eq 7). To elucidate the origin of the dependence on $\boldsymbol{n}$, we explicitly derive those covariance terms starting from the expressions for activity contributions $\boldsymbol{x}^{\text{loc}}$ and $\boldsymbol{x}^{\text{rec}}$ (Eqs 8, 9). (While equivalent to the general derivation provided in Methods 2, this calculation offers a more direct perspective on the origin of the dependence on $\boldsymbol{n}$.) Of the two covariance terms in Eq 17 that depend on $\boldsymbol{n}$, we focus on the first one; similar algebra applies to second one.

We start by introducing a compact notation for the signal obtained by filtering the stochastic process $\chi_i(t)$ with an exponential kernel of timescale $\Lambda$:

$$\hat{\chi}_i^{\Lambda} = \int_0^t \exp[\Lambda(t-v)]\chi_i(u)\,\mathrm{d}u. \tag{18}$$

Using this notation, the element $ij$ of the covariance term we seek to re-compute is given by:

$$\langle x_i^{\text{loc}} x_j^{\text{rec}} \rangle = \frac{k}{\lambda} m_j \sum_{k=1}^{N} n_k \left[ \langle \hat{\chi}_i^{-1} \hat{\chi}_k^{\lambda-1} \rangle - \langle \hat{\chi}_i^{-1} \hat{\chi}_k^{-1} \rangle \right] \tag{19}$$

where we used Eq 8 and Eq 9 to express $x_i^{\text{loc}}$ and $x_j^{\text{rec}}$, and we set for simplicity $U = I$. Observing that filtered processes $\hat{\chi}_i$ and $\hat{\chi}_k$ are uncorrelated when averaging with respect to different input realizations: $\langle \hat{\chi}_i^{\Lambda} \hat{\chi}_k^{\Lambda'} \rangle = 0$ for $i \neq k$ and every value of $\Lambda, \Lambda'$, we have that

$$\langle x_i^{\text{loc}} x_j^{\text{rec}} \rangle = \frac{k}{\lambda} m_j n_i \left[ \langle \hat{\chi}_i^{-1} \hat{\chi}_i^{\lambda-1} \rangle - \langle (\hat{\chi}_i^{-1})^2 \rangle \right]. \tag{20}$$

In the last equation, the dependence on $\boldsymbol{n}$ appears in the same form as in Eq 17.

We now paraphrase this derivation in words. For neuron $i$, the local activity term, $x_i^{\text{loc}}$, is given by a filtered version of the local stochastic input, $\chi_i$. In contrast, for neuron $j$, the recurrent activity term, $x_j^{\text{rec}}$, is expressed as a sum over *all* stochastic inputs $\chi_k$, each filtered and weighted by $m_j n_k$. When computing the cross-covariance between these two terms, only the stochastic input $k = i$ within $x_j^{\text{rec}}$ contributes a non-zero term. Consequently, the covariance between units $i$ and $j$ is proportional to $m_j n_i$.

This derivation highlights how the dependence of covariance on the connectivity vector $\boldsymbol{n}$ arises from activity comprising two components: one representing the integrated local stochastic input and the other reflecting integrated inputs to all neurons, aggregated through recurrent connectivity (Eq 7). In Methods 5, we discuss how these two activity components robustly emerge in continuous- and discrete-time RNNs. We also show that the local component takes a much simplified form in discrete-time RNNs characterized by extremely fast updates [38]. For these models, which are widely used in machine learning [39] and statistical applications [28], the activity covariance is expected to take a different functional form.

## 4.2. Covariance eigenvalues and eigenvectors

To quantify the statistical properties of activity and its dependence on connectivity vectors, we now compute the eigenvalues and eigenvectors of the covariance (Eq 17). Because of the properties of the identity matrix, the eigenvalues of $\Sigma$ are given by

$$\mu_i = \frac{1}{2}\left(1 + \mu_i^{\text{lr}}\right), \tag{21}$$

where $\{\mu_i^{\mathrm{lr}}\}_i$ are the eigenvalues of the low-rank covariance component

$$\Sigma^{\mathrm{lr}} = \frac{k}{2-\lambda} \left[ \boldsymbol{nm}^\top + \boldsymbol{mn}^\top + \frac{k}{1-\lambda} \boldsymbol{mm}^\top \right]. \tag{22}$$

$\Sigma^{\mathrm{lr}}$ has $N$–2 vanishing eigenvalues, resulting in $N$–2 eigenvalues for $\Sigma$ that are equal to the reference value $\mu_{\mathrm{ref}} = 1/2$. These eigenvalues correspond to eigenvectors, or PCs, along which activity is dominated by external inputs and receives no contribution from recurrent interactions. To compute the non-vanishing eigenvalues, we again follow [37]. A little algebra gives (see Methods 4):

$$\mu_\pm^{\mathrm{lr}} = \frac{k}{2(2-\lambda)} \left[ 2\rho_{mn} + \frac{k}{1-\lambda} \pm \sqrt{ \left( 2\rho_{mn} + \frac{k}{1-\lambda} \right)^2 + 4(1-\rho_{mn}^2) } \right] \tag{23}$$

from which $\mu_\pm$ can be computed via Eq 21. These eigenvalues correspond to eigenvectors, or PCs, along which activity is shaped by recurrent interactions. One can see from Eq 23 that $\mu_+^{\mathrm{lr}}$ and $\mu_-^{\mathrm{lr}}$ take values that are, respectively, always positive and negative. Therefore, $\mu_+$ and $\mu_-$ take values that are, respectively, larger and smaller than the reference value. This implies that the principal components associated with $\mu_+$ and $\mu_-$ identify directions along which activity is maximally amplified and maximally shrunk (Fig 3B).

These directions can be computed as the eigenvectors of $\Sigma^{\mathrm{lr}}$ associated with $\mu_\pm^{\mathrm{lr}}$. Similarly to the previous section, we set the ansatz:

$$\boldsymbol{v}_\pm = \gamma_\pm \boldsymbol{m} + \boldsymbol{n}. \tag{24}$$

A little algebra shows that $\gamma_\pm = \mu_\pm^{\mathrm{lr}}(2-\lambda)/k - \rho_{mn}$ (see Methods 4).

Combining these results, we conclude that the activity is isotropic, with amplitude equal to the reference value, across all dimensions orthogonal to the plane defined by the connectivity vectors $\boldsymbol{m}$ and $\boldsymbol{n}$. Within this plane, activity exhibits greater variance along one dimension (corresponding to the principal component associated with the largest covariance eigenvalue, PC1) and lower variance along the other (corresponding to the principal component associated with the smallest covariance eigenvalue, PC$N$). This results in covariance spectra that are largely flat (Fig 4A and 4D), but exhibit distinct values at the low and high ends. This profile contrasts with the gradually decaying spectra typically observed in networks with high-rank connectivity [18]. Therefore, networks with high- and low-rank connectivity driven by high-dimensional inputs can be distinguished based on the profile of their covariance spectra – an activity property that can be easily estimated from experimental data [40].

## 4.3. Dependence on connectivity vectors overlap

We analyze in detail the behaviour of the covariance eigenvalues and eigenvectors as a function of $\rho_{mn}$, the parameter that controls the level of symmetry of the recurrent connectivity matrix by specifying the overlap between the connectivity vectors $\boldsymbol{m}$ and $\boldsymbol{n}$. The network activity is stable for $\rho_{mn}$ values ranging between –1 and $1/k$, at which point the connectivity eigenvalue $\lambda$ reaches one. At one extreme, when $\boldsymbol{m}$ and $\boldsymbol{n}$ are strongly negatively correlated, the network functions as a one-dimensional auto-encoder [25,41,42], with recurrent connectivity generating strong negative feedback. At the other extreme, when $\boldsymbol{m}$ and $\boldsymbol{n}$ are strongly positively correlated, the network behaves as a line-attractor [43], where recurrent

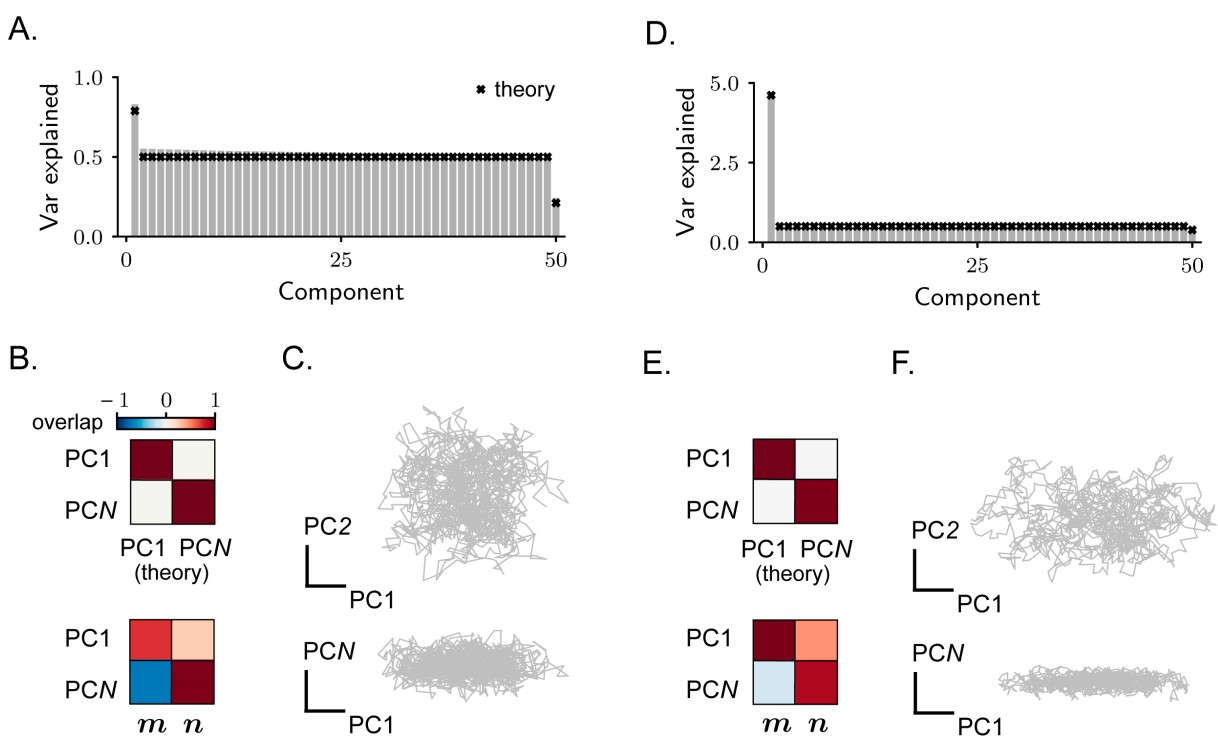

**Fig 4. Rank-one RNN receiving high-dimensional stochastic inputs. A–B–C.** Example of a simulated network with $\rho_{mn}$ = –0.5. In A: covariance spectrum. In B: overlap between two principal components (the strongest and the weakest) estimated from simulated activity and the theoretically-estimated vectors $\boldsymbol{v}_+$ and $\boldsymbol{v}_-$ (top), or vectors $\boldsymbol{m}$ and $\boldsymbol{n}$ (bottom). Overlaps are quantified via Eq 4. In C: simulated activity projected on two different pairs of PCs. **D–E–F.** Same as in A–B–C, example with $\rho_{mn}$ = 0.3. Note that, although the qualitative behaviour of activity in the two examples is similar, activity in the example network in A–B–C is overall higher dimensional.

connectivity mediates the integration of the external inputs by generating positive feedback that counteracts the leak dynamics.

We start by examining the case of strong positive overlap. Using $\rho_{mn} \to 1/k$ and $\lambda \to 1$, and keeping only the dominant terms, we have

$$\mu_\pm^{\mathrm{lr}} \simeq \frac{k}{2}\left[\frac{k}{1-\lambda} \pm \left|\frac{k}{1-\lambda}\right|\right],$$

(25)

implying that $\mu_+^{\mathrm{lr}} \to +\infty$ and $\mu_-^{\mathrm{lr}} \to 0$ or, equivalently, $\mu_+ \to +\infty$ and $\mu_- \to 1/2$ (Fig 3C). Therefore, activity is characterized by a diverging component along $\boldsymbol{v}_+$, reflecting the noise integration. Along all other directions, including $\boldsymbol{v}_-$, the amplitude of activity is fixed to the reference value. Therefore, the dimensionality of activity approaches one (Fig 3D). The principal component $\boldsymbol{v}_+$ is primarily aligned with the connectivity vector $\boldsymbol{m}$, as indicated by the fact that $\gamma_+ \to \infty$ (Fig 3E). However, $\boldsymbol{v}_+$ still retains a nonzero overlap with $\boldsymbol{n}$, since $\boldsymbol{m}$ and $\boldsymbol{n}$ are not orthogonal. Simulated activity for networks with positive overlap, operating close to this regime, is shown in Fig 4D, 4E and 4F.

We now examine the opposite scenario of strong negative overlap. As shown in Methods 4, using $\rho_{mn} \to -1$ and $\lambda \to -k$, the value of $\mu_+$ becomes equal to the reference value. The value of $\mu_-$, instead, becomes significantly smaller, and approaches zero as the strength of recurrent connectivity $k$ is increased (Fig 3C). Therefore, activity is characterized by a strongly reduced component along $\boldsymbol{v}_-$. Along all other directions, including $\boldsymbol{v}_+$, the amplitude of activity is fixed

to the reference value. As the suppression along $v_-$ has a limited impact on activity (in contrast to the previous case with positive overlap, where the expansion along $v_+$ is unbounded), the dimensionality of activity is high (Fig 3D). The principal component $v_-$ is mostly aligned with the connectivity vector $n$, as it can be shown that the value of $\gamma_-$ approaches zero for strong recurrent connectivity (Fig 3E). Also in this case, $v_-$ retains a nonzero overlap with $m$, since $m$ and $n$ are strongly aligned. Simulated activity for networks with negative overlap, operating close to this regime, is shown in Fig 4A, 4B and 4C.

Finally, for connectivity matrices characterized by intermediate or vanishing overlaps, activity is characterized by intermediate properties: the values of $\mu_+$ and $\mu_-$ are respectively larger and smaller than the reference values (Fig 3C). Therefore, activity is characterized by an enhanced component along $v_+$ and a reduced one along $v_-$; dimensionality takes intermediate values (Fig 3D). The principal components $v_+$ and $v_-$ have non-vanishing components along both connectivity vectors $m$ and $n$; $v_+$ retains a stronger overlap with $m$, while $v_-$ a stronger overlap with $n$ (Fig 3E).

To summarize, under high-dimensional stochastic inputs, recurrent connectivity contributes to activity covariance by structuring it along two dimensions. In the limiting cases of strongly positive or negative overlap between the connectivity vectors $m$ and $n$ – corresponding to attractor [43] and autoencoder [41] models – the effect of recurrence simplifies to a single dominant dimension. The qualitative features of activity differ markedly between these two regimes: for positive overlap, activity is strongly amplified along the direction most correlated with $m$, resulting in low dimensionality; for negative overlap, activity is suppressed along the direction most correlated with $n$, leading to high dimensionality. Substantial differences in activity dimensionality have also been reported in experimental studies, with low-dimensional activity often observed in frontal and associative cortices [5,6], but not necessarily in sensory areas [40]. Our findings indicate that, within the context of models with low-rank connectivity, these differences can be reconciled by positing that associative regions are mostly characterized by positive connectivity overlap (leading to attractor-like dynamics), while sensory regions exhibit more negative overlap (leading to autoencoder-like activity).

## 5. Stochastic activity in higher-rank networks

Our analysis so far has focused on connectivity matrices of unit rank, which have been extensively studied before in the context of simplified external inputs [8,13]. We have shown that introducing inputs with more complex structure – such as high-dimensional ones – qualitatively alters the geometrical properties of the resulting activity, both in terms of its dimensionality and in the functional relationship between the connectivity vectors and the activity covariance. A summary of these results is reported in Table 1.

In this section, we discuss how results generalize to matrices of higher rank. Those matrices can be written as

$$W = k \sum_{r=1}^{R} m^r n^{r\top} \tag{26}$$

where once again $k$ is a finite scalar and connectivity vectors $m^r$ and $n^r$ for $r = 1, \dots, R < N$ are unit norm. In particular, we focus on a subset of low-rank matrices (see Methods 6) for which the non-zero eigenvalues are given by $\lambda^r = k\rho_{m^r n^r}$, each corresponding to an eigenvector $m^r$. (Here and in the following, the superscript $r$ is an index referring to the rank expansion of $W$, Eq 26, and does not denote an exponent.)

**Table 1**. **Summary of the main results obtained for recurrent neural networks with rank-one connectivity.** In the second column, we report the directions defining the part of the covariance matrix (Eq 11) that originates from recurrent activity. (Note that also input directions contribute, via the term $\bar{\Sigma}_{\text{inp}}$.) In the fourth column, the structure of connectivity vectors determines the precise value taken by dimensionality. For intermediate input dimensionality, covariance spectra have not been formally analyzed. Simulations suggest that, among the $C + 1$ eigenvalues that do not vanish, $C - 2$ are approximately fixed to the reference value, while three depend on connectivity.

| Input dimensionality | Covariance directions from recurrence | Covariance eigenvalues | Activity dimensionality |
|---|---|---|---|
| 1 (one-dimensional) | $\boldsymbol{m}$, $\boldsymbol{u}$ | Two depend on connectivity, remaining ones vanish | from 1 to 2 |
| $N$ (high-dimensional) | $\boldsymbol{m}$, $\boldsymbol{n}$ | Two depend on connectivity, remaining ones fixed | from 1 to $N$ |
| $C$ (intermediate) | $\boldsymbol{m}$, $\sum_c \rho_{nu^c} \boldsymbol{u}^c$ | Not discussed | from 1 to $C + 1$ |

For these matrices, the propagator can again be computed in closed form (Eq 89, see Methods 6), and therefore activity can be computed analytically. In analogy with Eq 7, we can write:

$$\boldsymbol{x}(t) = \boldsymbol{x}^{\text{loc}}(t) + \sum_{r=1}^{R} \boldsymbol{x}^{\text{rec},r}(t) \tag{27}$$

where the component $\boldsymbol{x}^{\text{loc}}$ is given by Eq 8, and each component $\boldsymbol{x}^{\text{rec},r}$ is given by Eq 9, with $\boldsymbol{m}$, $\boldsymbol{n}$, and $\lambda$ replaced, respectively, by $\boldsymbol{m}^r$, $\boldsymbol{n}^r$, and $\lambda^r$. Therefore, increasing the rank of the synaptic connectivity primarily increases the number of independent activity components that are generated by recurrent dynamics.

What is the impact of this increase on the geometrical properties of activity? Similarly to the rank-one case, we can compute the covariance matrix, which is given by (see Methods 7):

$$\Sigma = \frac{1}{2} \left\{ \bar{\Sigma}_{\text{inp}} + \sum_{r=1}^{R} \frac{k}{2 - \lambda^r} \left[ \boldsymbol{d}^r \boldsymbol{m}^{r\top} + \boldsymbol{m}^r \boldsymbol{d}^{r\top} \right] + \sum_{r=1}^{R} \sum_{r'=1}^{R} \frac{\sigma^{rr'} k^2 (4 - \lambda^r - \lambda^{r'})}{(2 - \lambda^r - \lambda^{r'})(2 - \lambda^r)(2 - \lambda^{r'})} \boldsymbol{m}^r \boldsymbol{m}^{r'\top} \right\} \tag{28}$$

where we defined $\boldsymbol{d}^r = \bar{\Sigma}_{\text{inp}} \boldsymbol{n}^r$ and $\sigma^{rr'} = \boldsymbol{n}^{r\top} \bar{\Sigma}_{\text{inp}} \boldsymbol{n}^{r'}$. To inspect the properties of this matrix, we once again focus on the two limiting cases, corresponding to one- and high-dimensional stochastic inputs.

For one-dimensional inputs, Eq 28 becomes:

$$\Sigma = \frac{1}{2} \left\{ \boldsymbol{u}\boldsymbol{u}^\top + \sum_{r=1}^{R} \frac{\rho_{n^r u} k}{2 - \lambda^r} \left[ \boldsymbol{u}\boldsymbol{m}^{r\top} + \boldsymbol{m}^r \boldsymbol{u}^\top \right] + \sum_{r=1}^{R} \sum_{r'=1}^{R} \frac{\rho_{n^r u} \rho_{n^{r'} u} k^2 (4 - \lambda^r - \lambda^{r'})}{(2 - \lambda^r - \lambda^{r'})(2 - \lambda^r)(2 - \lambda^{r'})} \boldsymbol{m}^r \boldsymbol{m}^{r'\top} \right\}. \tag{29}$$

This matrix has a low-rank structure, spanned by $R + 1$ vectors, and therefore its rank is at most $R + 1$. As a result, the activity is low-dimensional and confined to the hyperplane spanned by the input vector $\boldsymbol{u}$ and the connectivity vectors $\boldsymbol{m}$. As in rank-one networks, the activity in this input regime qualitatively resembles the patterns described in earlier work on deterministic low-rank RNNs [13].

For high-dimensional inputs, we have instead

$$\Sigma = \frac{1}{2} \left\{ I + \sum_{r=1}^{R} \frac{k}{2 - \lambda^r} \left[ \boldsymbol{n}^r \boldsymbol{m}^{r\top} + \boldsymbol{m}^r \boldsymbol{n}^{r\top} \right] + \sum_{r=1}^{R} \sum_{r'=1}^{R} \frac{\rho_{n^r n^{r'}} k^2 (4 - \lambda^r - \lambda^{r'})}{(2 - \lambda^r - \lambda^{r'})(2 - \lambda^r)(2 - \lambda^{r'})} \boldsymbol{m}^r \boldsymbol{m}^{r'\top} \right\}. \tag{30}$$

This covariance matrix has high rank, as it consists of the sum of a full-rank term (the first term on the right-hand side) and a low-rank term with maximal rank $2R$. This observation generalizes the results obtained for rank-one networks: in the presence of high-dimensional stochastic inputs, network activity becomes potentially high-dimensional, with recurrent connectivity shaping activity along a number of dimensions equal to twice the rank of the connectivity. This arises because activity is structured along both sets of connectivity vectors, $\boldsymbol{m}$ and $\boldsymbol{n}$.

In Fig 5, we analyze in detail rank-two networks by focusing on two specific connectivity parametrizations. In Fig 5B and 5C, we fix the overlap $\rho_{m^1 m^2}$ and $\rho_{n^1 n^2}$ to zero, while varying $\rho_{m^1 n^1}$ and $\rho_{m^2 n^2}$. This case corresponds to rank-one components in the connectivity (Eq 26) that operate on orthogonal subspaces. The eigenvalues of the recurrent connectivity in this parametrization vary together with $\rho_{m^1 n^1}$ and $\rho_{m^2 n^2}$. In Fig 5D and 5E, instead, we fix $\rho_{m^1 n^1}$ and $\rho_{m^2 n^2}$ to zero, while varying $\rho_{m^1 m^2}$ and $\rho_{n^1 n^2}$. This parametrization results in a connectivity matrix with vanishing eigenvalues, and arbitrary correlations among rank-one components.

As expected, for matrices obeying both the first (Fig 5B) and the second (Fig 5D) parametrizations, recurrent connectivity impacts the covariance by modifying the value of four eigenvalues. Among these four eigenvalues, two of them are decreased with respect to the reference value $\mu_{\text{ref}}$, while two are increased. Similarly to the rank-one case, activity is therefore compressed along two directions and stretched along two others (Fig 5F). The activity PCs that correspond to these transformations are linear combinations of the four connectivity vectors (Fig 5G).

In Fig 5C and 5E, we finally leverage the analytical expressions for the covariance eigenvalues to compute dimensionality. We find that activity generally exhibits relatively high dimensionality across most of the parameter space. Low-dimensional activity emerges only in regions where one or both rank-one components of the connectivity are characterized by a large internal overlap $\rho_{m^r n^r}$, leading to large and positive eigenvalues. In these cases, recurrent connectivity generates positive feedback resulting in a temporal integration of the external inputs, similarly to the rank-one case discussed in Fig 4D and 4F.

## 6. Amplification in excitatory-inhibitory networks

As a final application of major biological relevance, we consider a circuit consisting of one excitatory (E) and one inhibitory (I) unit. The connectivity is given by

$$W = w \begin{pmatrix} 1 & -g \\ 1 & -g \end{pmatrix} \tag{31}$$

where $w$ and $g$ are two non-negative scalars. This matrix is rank-one and can be rewritten in the form of Eq 3 by setting

$$\begin{aligned} \boldsymbol{m} &= \frac{1}{\sqrt{2}}(1, 1) \\ \boldsymbol{n} &= \frac{1}{\sqrt{g^2 + 1}}(1, -g) \end{aligned} \tag{32}$$

as well as $k = w\sqrt{2(g^2 + 1)}$. The connectivity vector $\boldsymbol{m}$ therefore corresponds to the *sum* direction $(1, 1)/\sqrt{2}$, expressing co-modulation of the E-I units, while the connectivity vector $\boldsymbol{n}$ is

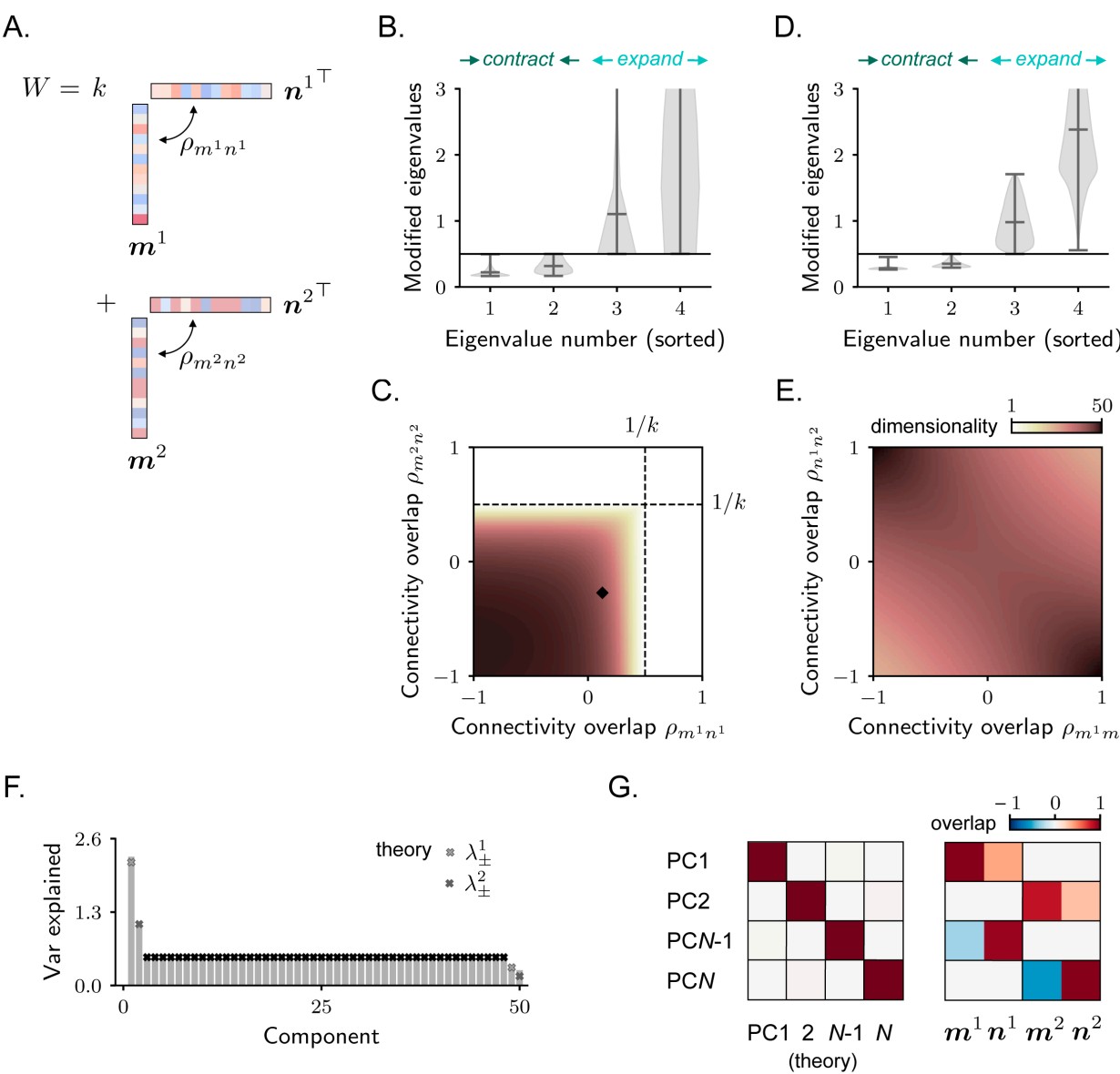

**Fig 5. Stochastic activity in rank-two recurrent neural networks. A.** Rank-two connectivity. **B.** Eigenvalues of the covariance matrix that are different than the reference value $\mu_{\text{ref}}$. As connectivity is rank-two, four eigenvalues are perturbed; we sort them in ascending order. Violin plots show the distribution of perturbed eigenvalues for different values of the parameters $\rho_{m^1 n^1}$ and $\rho_{m^2 n^2}$. Note that, for all sets of parameters, two eigenvalues are increased and two decreased with respect to $\mu_{\text{ref}}$. **C.** Dimensionality as a function of $\rho_{m^1 n^1}$ and $\rho_{m^2 n^2}$. The black dashed lines indicate the parameter values for which dynamics become unstable. The tiny black square indicates the parameter values that are used for simulations in **F–G**. In both **B** and **C**, we keep the values of $\rho_{m^1 m^2}$ and $\rho_{n^1 n^2}$ fixed to zero (see Methods 7). **D–E.** Same as for **B–C**, but for a different parametrization, where we keep $\rho_{m^1 n^1}$ and $\rho_{m^2 n^2}$ fixed to zero. **F–G.** Example of a simulated network, parameters indicated in C. In F: covariance spectrum. In G: overlap between four selected principal components (the strongest and the weakest) estimated from simulated activity and the theoretically-estimated covariance eigenvectors (left) and the connectivity vectors (right). Overlaps are quantified via Eq 4. The theoretical expressions for this case are reported in Methods 7.

approximately aligned with the *difference* direction $(1, -1)/\sqrt{2}$, capturing opposite-sign modulations of E and I [44]. The non-zero eigenvalue of the connectivity is given by $\lambda = w(1 - g)$, which is negative in the inhibition-dominated regime ($g > 1$).

A key property of E-I connectivity in cortical circuits is its ability to amplify inputs while preserving fast dynamics [44]. This is achieved through a mechanism known as non-normal amplification, arising in circuits with non-symmetric connectivity matrices and non-orthogonal eigenvectors. The stochastic setting we consider here provides a natural framework for studying input amplification [16,45]. We therefore leverage our mathematical framework to analyze stochastic dynamics in the rank-one E-I circuit (Eq 31), with a particular focus on amplification and its dependence on the geometry of external inputs.

We begin by considering the case in which E and I units receive uncorrelated stochastic inputs (high-dimensional inputs, Fig 6A). The covariance matrix (Eq 17) has two eigenvalues, illustrated in Fig 6B. The second eigenvalue is significantly smaller than the first (note the different color scales), indicating that activity is approximately one-dimensional, with nearly all variance concentrated along the first principal component. To characterize PC1 (Eq 24), we compute its overlap with the sum and difference directions (Fig 6C). Consistent with previous work, we find that PC1 is much more strongly aligned with the sum mode than with the difference mode [44].

The amplitude of the dominant covariance eigenvalue provides a measure of input amplification. Fig 6B shows that the degree of input amplification performed by the recurrent connectivity depends on the parameters $w$ and $g$, quantifying the total connectivity strength and the relative strength of inhibition over excitation, respectively. In particular, amplification is large in a parameter region adjacent to the unstable region, where the connectivity eigenvalue $\lambda$ exceeds the stability threshold, and the input noise is integrated via slow dynamics [16]. However, relatively large amplification can also occur in an adjacent region where the connectivity eigenvalue $\lambda$ is negative (Fig 6D), and non-normal amplification mechanisms are at play. This region corresponds to values of $g$ slightly greater than one (weak inhibition dominance) and large $w$ (strong excitation and inhibition).

To understand how variance amplification is linked to the geometry of external inputs, we next consider the case of one-dimensional inputs (Fig 6E). We construct a family of input vectors, $u$, each characterized by a different relative strength of input to E versus I. As before, we compute the covariance eigenvalue and eigenvectors as functions of the parameters $w$ and $g$. As in the high-dimensional input configuration, we find that activity is often concentrated along PC1, expressing co-modulatory variance in the E and I units (sum direction, Fig 6H). Crucially, however, the amplitude of input amplification and its dependence on the connectivity parameters vary with the input vector direction (Fig 6F and 6G). For input vectors strongly aligned with the difference direction (or connectivity vector $n$), input amplification can be large; in particular, it reaches a maximum for large values of $w$ and values of $g$ slightly greater than one, as in the previous case. Instead, for inputs strongly aligned with the sum direction (or connectivity vector $m$), input amplification is small and largely independent of the connectivity parameters. This can be understood from Eq 13, as this configuration corresponds to very small values of $\rho_{nu}$, leading to minimal engagement of recurrent connectivity in processing the inputs. Note that along the vertical slices of the plots in Fig 6F and 6G, the eigenvalues of the synaptic connectivity remain fixed. Thus, the observed modulation in variance as a function of input direction must be attributed to the non-normal properties of the connectivity.

The dependence of variance on input direction described here is broadly consistent with previous findings in deterministic networks [12,44]. In particular, for fixed connectivity parameters, the input direction that leads to maximal amplification is close to the difference direction [44], but is not identical – consistent with the results from [12].

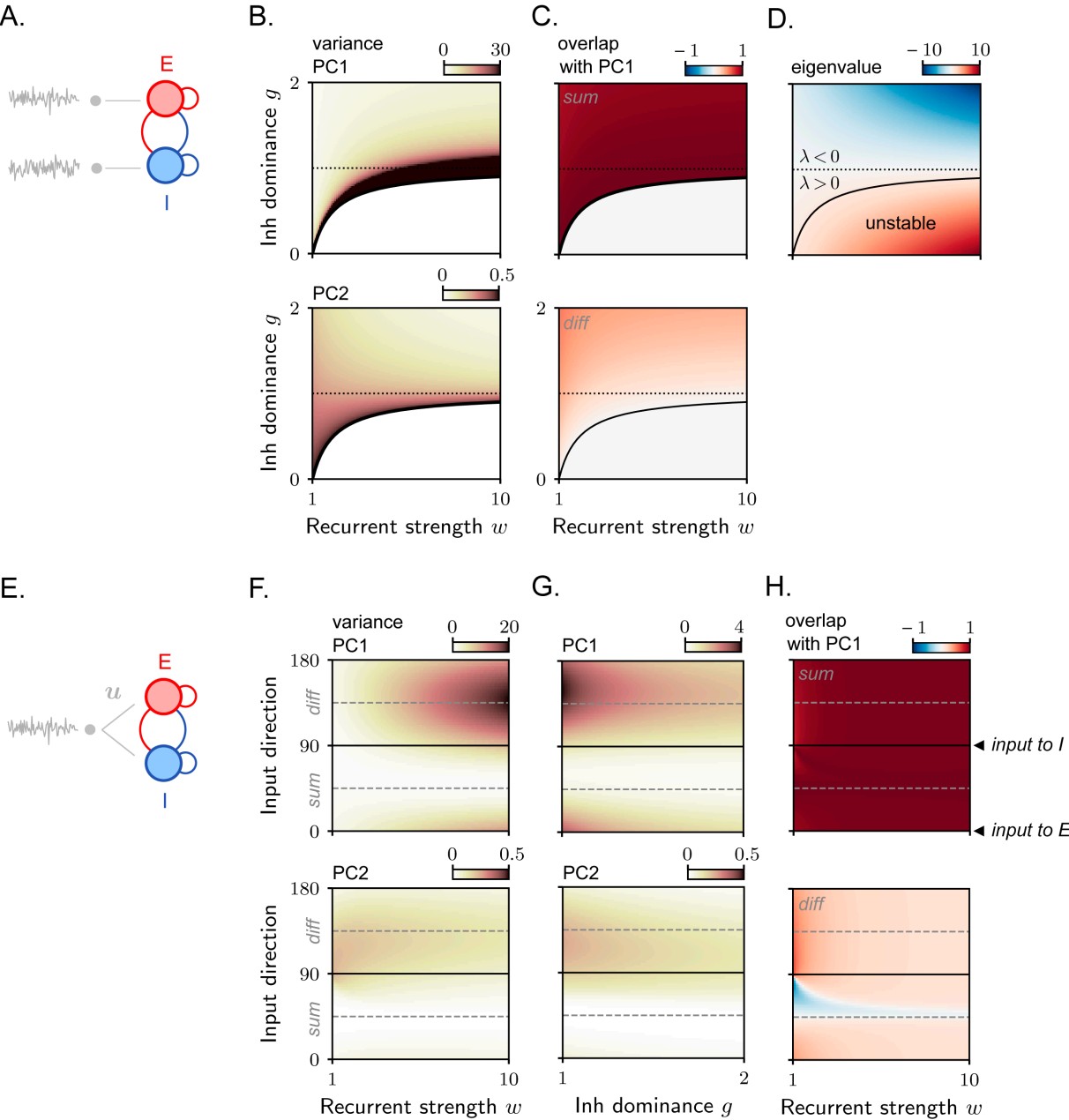

**Fig 6. Stochastic activity in a rank-one excitatory-inhibitory circuit. A.** E-I circuit with high-dimensional inputs. **B.** Variance explained by PC1 (top) and PC2 (bottom) as a function of the overall recurrent connectivity strength $w$ and the relative dominance of inhibition $g$. In B–C–D, the black solid line separates the regions for which the non-zero eigenvalue $\lambda$ is larger or smaller than one. The black dotted line separates the regions for which the non-zero eigenvalue $\lambda$ is larger or smaller than zero. Note the different color scales in the top and bottom plots. **C.** Overlap between PC1 and the sum (top) and diff (bottom) directions. **D.** Non-zero eigenvalue of the synaptic connectivity matrix $\lambda = w(1 - g)$. **E.** E-I circuit with one-dimensional inputs. **F.** Variance explained by PC1 (top) and PC2 (bottom) as a function of the overall recurrent connectivity strength $w$ and the direction of the input vector $\boldsymbol{u}$. The input direction is parametrized by an angle $\theta$ (see Methods 8), so that $\theta = 0°$ (resp. 90°) correspond to inputs entering only E (resp. I), while $\theta = 45°$ (resp. 135°) corresponds to inputs aligned with the sum (resp. diff) direction. **G.** Variance explained by PC1 (top) and PC2 (bottom) as a function of the relative dominance of inhibition $g$ and the direction of the input vector $\boldsymbol{u}$. **H.** Overlap between PC1 and the sum (top) and diff (bottom) direction.

## Discussion

The rapid advancement of experimental techniques over the past decade has led to extensive datasets characterizing single-neuron activity across large populations and brain regions [40, 46], as well as increasingly detailed maps of their synaptic connectivity [47,48]. A key challenge is to understand how these two levels of organization relate to each other during brain function. Theoretical models play a fundamental role in this effort, by helping to synthesize hypotheses about the relationship between connectivity and activity.

In this work, we investigated this relationship in linear RNNs with low-rank connectivity under the influence of stochastic external inputs. We analyzed activity across different choices of external input dimensionality, ranging from low to high. Our findings show that the dimensionality of emergent activity is strongly influenced by input dimensionality, with low-dimensional activity arising systematically only when the inputs themselves are low-dimensional. For high-dimensional inputs, the dimensionality of activity can be high, and depends on the statistical properties of the connectivity. Specifically, low-dimensional activity only emerges in networks with connectivity matrices that have large positive eigenvalues (Fig 4D), where recurrent interactions induce unbounded amplification along the directions spanned by the connectivity vectors. In contrast, activity is maximally high-dimensional in networks with negative eigenvalues (Fig 4A), where recurrent interactions predominantly have a suppressive effect.

Recent parallel work by Wan and Rosenbaum [10] analyzed input responses in large low-rank networks with strong recurrent connectivity. In that framework, the entries of the low-rank connectivity scale with network size as $1/\sqrt{N}$, in contrast to our setting where the scaling is $1/N$. This difference has an important implication: the connectivity in [10] only yields stable dynamics in the case of negative eigenvalues, whose magnitude typically diverges with the network size. Within that regime, the authors report high-dimensional activity and suppression of inputs along the directions defined by the connectivity vectors. Our results are qualitatively consistent with these findings. In our framework, networks with negative overlap $\rho_{mn}$ exhibit negative eigenvalues; as shown in Fig 3C and 3D, the corresponding dynamics are primarily suppressive, giving rise to high-dimensional activity. Compared to [10], our framework accommodates a broader range of scenarios. In particular, our analysis reveals that even for positive eigenvalues, some form of suppression can still be observed: while activity is amplified along certain directions, it is concurrently suppressed along others, resulting in intermediate dimensionality.

Our analysis uncovered a second fundamental difference between the low- and high-dimensional input regimes. In networks receiving low-dimensional inputs, the structure of the activity covariance depends on connectivity solely through the set of vectors **m**. In contrast, in networks receiving high-dimensional inputs, covariance explicitly depends on both sets of connectivity vectors, **m** and **n**. These two sets of vectors define the 2$R$-dimensional hyperplane in the activity space where activity distribution is stretched along some directions and shrunk along others. The dependence of activity covariance on the connectivity vectors **m** has already been reported in previous work on low-rank RNNs [8,33,34]. This dependence arises because, due to the low-rank structure of recurrent connectivity, the contribution to activity that stems from recurrent interactions (here, Eq 9) is always constrained to align with **m**. In contrast, the dependence of activity covariance on the connectivity vector **n** has not been previously emphasized (but see [12,34]). We show that this dependence emerges only when the inputs are high-dimensional – a setup that is relevant for both biological and artificial networks, but has not been mathematically analyzed in detail before.

Previous work has demonstrated that the direction of connectivity vectors $m$ can be robustly inferred from activity traces – whether obtained from simulations or experimental data – using simple statistical techniques, such as principal component analysis [8,34]. In contrast, reliably estimating the direction of vectors $n$ from data has generally been considered a more challenging problem [34]. In this work, we show that for the activity covariance matrix to explicitly encode information about $n$, the RNN dynamics must be driven by high-dimensional external inputs. This finding suggests that, in the high-dimensional input regime, inference algorithms may more easily and accurately reconstruct the directions of these vectors. This idea is supported by a simple intuition: uniquely identifying the input direction that generates recurrent activity along $m$ (which is precisely the role of $n$) requires observing the system's response across multiple input directions. This hypothesis is further consistent with recent work by Quian et al. [49], which demonstrated that connectivity vectors can be reliably estimated in a teacher-student setup involving high-dimensional stochastic inputs as long as activity traces from all network units are available. That study focused on a specific class of low-rank RNNs, where the eigenvalues of the connectivity matrix vanish, and mostly examined the case of very strong connectivity weights, which lead to approximately low-dimensional activity even in response to high-dimensional inputs. In contrast, here we considered generic low-rank matrices, with a particular focus on rank-one structures, and we provided a detailed analysis of how arbitrary input and connectivity vector geometries affect the emergent activity.

The analysis in [49] also highlights an important challenge: available algorithms for inferring low-rank dynamics from activity datasets often struggle when applied to subsampled neural data. Subsampling leads to an overestimation of symmetry in the connectivity matrix and, in general, poor performance in reconstructing non-normal dynamics [50]. Developing algorithms that can handle such challenges, and provide an effective and scalable estimate of both connectivity vectors regardless of their overlap, represents an important avenue for future research. Recently, novel algorithms for inferring RNN models from neural data have been proposed, which leverage the probabilistic structure of activity that emerges from stochastic inputs [30,51]. Investigating the performance of these algorithms under subsampling and other aversive conditions represents an important step for future research. In the context of those studies, our analysis provides a solid mathematical framework on which these statistical techniques can be benchmarked.

Another domain where our results may open new avenues is the study of learning in biological and artificial recurrent neural networks. Low-rank connectivity matrices are particularly relevant in this context, as synaptic updates often take a low-rank form [23,31], and in some cases, the final learned connectivity also retains a low-rank structure [26,27]. Most learning algorithms adjust synaptic connectivity based on estimates of network activity and its covariance structure [22,23,31,52]. In this work, we have showed that both activity and covariance depend sensitively on the dimensionality and geometry of stochastic inputs. As a result, in plastic recurrent networks, different noise structures in the inputs are expected to produce not only distinct activity patterns but also fundamentally different connectivity profiles. Consistent with these predictions, recent work has demonstrated that, in recurrent networks trained via gradient descent, the presence of stochastic inputs can qualitatively alter the solution found by the learning algorithm [25]. For instance, noise can bias learning toward configurations that better align with experimental observations [53], or induce additional low-rank connectivity components that suppress variability along task-relevant directions [42]. However, this study did not systematically investigate how the amplitude and geometry of input noise shape the learned connectivity. Our framework offers several predictions about this relationship, which remain to be tested and explored in future work. For example,

since the rank of the learned connectivity is linked to the dimensionality of network activity, we expect it to scale with the dimensionality of external stochastic inputs. Moreover, we have seen that any learned low-rank connectivity term effectively suppresses variability along specific directions, which could be exploited by the learning algorithm for robust decoding of information. We finally remark that, although trained networks often rely on non-linear dynamics, our linear framework remains broadly applicable, as linearization around fixed points is a common and powerful strategy for analyzing such systems [26,54].

As a straightforward application of biological relevance, we considered a two-dimensional circuit consisting of one excitatory and one inhibitory unit, with rank-one connectivity. Although this system and its input amplification properties have been the focus of several studies [12,16,44], to our knowledge, a systematic analysis for different input structures in the stochastic setting has not been conducted before. Our analysis highlights that, in these systems, large non-normal amplification can be observed provided that both excitation and inhibition are strong and connectivity is weakly inhibition-dominated. This range of parameters maximizes amplification for both low- [44] and high-dimensional [16] external inputs. However, as expected from Eq 13, amplification occurs for low-dimensional inputs only when input vectors are specifically oriented with respect to the connectivity vectors. In particular, non-normal amplification is maximized when the input vector modulates the excitatory and inhibitory units in opposition (difference direction) [44]. Non-normal amplification takes intermediate values for inputs targeting only the excitatory or inhibitory units – a scenario relevant to the study of optogenetic perturbations in cortical circuits [50,55]. In particular, amplification is greater for inputs targeting only the excitatory unit than for those targeting only the inhibitory unit (Fig 6F and 6G; compare the horizontal black lines at 90° and 180°).

Finally, our framework opens new avenues for studying how trial-to-trial variability in brain circuits relates to the underlying anatomical structure. Stochastic inputs have long been used in recurrent neural networks to model fluctuations in brain activity [2,16,17,56–58]. However, most studies have focused on models with simple connectivity structures, such as fully random networks or those incorporating local connectivity motifs. Our work extends this approach to low-rank connectivity structures, which are global and more naturally linked to network computations [26,33]. In particular, low-rank RNNs have recently been used to model mesoscopic cortical circuits spanning multiple brain areas [59,60]. Experimental studies indicate that these circuits exhibit structured trial-to-trial variability, with cross-covariances differing in dimensionality when computed within versus across areas [61]. Our work provides the mathematical tools needed to analyze variability in these models, offering a foundation for refining them through direct comparison with experimental data.

## Acknowledgments

We would like to thank Giulio Bondanelli and Srdjan Ostojic for their feedback on the manuscript.

## Author contributions

**Conceptualization:** Francesca Mastrogiuseppe.

**Formal analysis:** Francesca Mastrogiuseppe.

**Funding acquisition:** Francesca Mastrogiuseppe, Joana Carmona, Christian K Machens.

**Investigation:** Francesca Mastrogiuseppe.

**Methodology:** Francesca Mastrogiuseppe.

**Software:** Francesca Mastrogiuseppe.

**Writing – original draft:** Francesca Mastrogiuseppe.

**Writing – review & editing:** Francesca Mastrogiuseppe, Joana Carmona, Christian K Machens.

## Methods

### 1. Propagator of rank-one matrices

We derive here the expression (Eq 6) for the propagator operator $\exp[(W - I)t]$ for a connectivity matrix $W$ of rank one as defined in the main text. We follow [12] and start observing that, since $W$ and $I$ commute,

$$\exp[(W - I)t] = \exp(Wt)\exp(-t). \tag{33}$$

Moreover, using $\lambda = k\rho_{mn}$, we have:

$$
\begin{aligned}
\exp(Wt) &= I + kt\, \boldsymbol{mn}^\top + \frac{(kt)^2}{2!}\, \boldsymbol{mn}^\top \boldsymbol{mn}^\top + \frac{(kt)^3}{3!}\, \boldsymbol{mn}^\top \boldsymbol{mn}^\top \boldsymbol{mn}^\top \dots \\
&= I + \frac{1}{\rho_{mn}}\left[(k\rho_{mn}t)\, \boldsymbol{mn}^\top + \frac{(k\rho_{mn}t)^2}{2!}\, \boldsymbol{mn}^\top + \frac{(k\rho_{mn}t)^3}{3!}\, \boldsymbol{mn}^\top \dots \right] \\
&= I + \frac{\exp(\lambda t) - 1}{\lambda} k\, \boldsymbol{mn}^\top
\end{aligned} \tag{34}
$$

Finally:

$$\exp[(W - I)t] = \exp(-t)\left[I + \frac{\exp(\lambda t) - 1}{\lambda} k\, \boldsymbol{mn}^\top\right]. \tag{35}$$

Note that this expression is only valid for $\lambda \neq 0$. When $\lambda = 0$, we have that

$$\exp(Wt) = I + k\, \boldsymbol{mn}^\top t \tag{36}$$

(all the higher-order terms in the power expansion vanish, being multiplied by $\lambda$). Therefore:

$$\exp[(W - I)t] = \exp(-t)\left[I + k\, \boldsymbol{mn}^\top t\right]. \tag{37}$$

### 2. Covariance derivation

From Eq 5, it follows that the mean activity, $\langle \boldsymbol{x}(t)\rangle$, vanishes in the stationary state. Consequently, the time-resolved covariance matrix can be expressed as:

$$
\begin{aligned}
\Sigma(t, s) &= \left\langle \left(\int_0^t \exp[(W - I)(t - u)]U\boldsymbol{\chi}(u)\,\mathrm{d}u\right)\left(\int_0^s \exp[(W - I)(s - v)]U\boldsymbol{\chi}(v)\,\mathrm{d}v\right)^\top \right\rangle \\
&= \int_0^t \int_0^s \exp[(W - I)(t - u)]\, \bar{\Sigma}_{\text{inp}}\, \exp[(W - I)(s - v)]^\top \delta(u - v)\,\mathrm{d}u\,\mathrm{d}v.
\end{aligned} \tag{38}
$$

 

Substituting the expression for the propagator (Eq 6), we see that covariance factorizes into four terms:

$$\Sigma(t,s) = \bar{\Sigma}_{\text{inp}} \int_0^t \int_0^s \exp[-(t+s-u-v)]\,\delta(u-v)\,\mathrm{d}u\,\mathrm{d}v$$

$$+ \frac{k}{\lambda}\bar{\Sigma}_{\text{inp}}\boldsymbol{nm}^\top \int_0^t \int_0^s \exp[-(t+s-u-v)]\,\{\exp[\lambda(s-v)]-1\}\delta(u-v)\,\mathrm{d}u\,\mathrm{d}v$$

$$+ \frac{k}{\lambda}\boldsymbol{mn}^\top\bar{\Sigma}_{\text{inp}} \int_0^t \int_0^s \exp[-(t+s-u-v)]\,\{\exp[\lambda(t-u)]-1\}\delta(u-v)\,\mathrm{d}u\,\mathrm{d}v \qquad (39)$$

$$+ \frac{k^2}{\lambda^2}\boldsymbol{mn}^\top\bar{\Sigma}_{\text{inp}}\,\boldsymbol{nm}^\top \int_0^t \int_0^s$$

$$\exp[-(t+s-u-v)]\,\{\exp[\lambda(t-u)]-1\}\,\{\exp[\lambda(s-v)]-1\}\delta(u-v)\,\mathrm{d}u\,\mathrm{d}v.$$

This expression can be re-written as

$$\Sigma(t,s) = c_1(t,s)\,\bar{\Sigma}_{\text{inp}} + c_2(t,s)\,\bar{\Sigma}_{\text{inp}}\boldsymbol{nm}^\top + c_3(t,s)\,\boldsymbol{mn}^\top\bar{\Sigma}_{\text{inp}} + c_4(t,s)\,\boldsymbol{mn}^\top\bar{\Sigma}_{\text{inp}}\,\boldsymbol{nm}^\top \qquad (40)$$

by defining:

$$c_1 = \int_0^t \int_0^s \exp[-(t+s-u-v)]\,\delta(u-v)\,\mathrm{d}u\,\mathrm{d}v$$

$$c_2 = \frac{k}{\lambda} \int_0^t \int_0^s \exp[-(t+s-u-v)]\,\{\exp[\lambda(s-v)]-1\}\delta(u-v)\,\mathrm{d}u\,\mathrm{d}v$$

$$c_3 = \frac{k}{\lambda} \int_0^t \int_0^s \exp[-(t+s-u-v)]\,\{\exp[\lambda(t-u)]-1\}\delta(u-v)\,\mathrm{d}u\,\mathrm{d}v$$

$$c_4 = \frac{k^2}{\lambda^2} \int_0^t \int_0^s \exp[-(t+s-u-v)]\,\{\exp[\lambda(t-u)]-1\}\,\{\exp[\lambda(s-v)]-1\}\delta(u-v)\,\mathrm{d}u\,\mathrm{d}v. \qquad (41)$$

Working out the integrals on the right-hand side yields the final expression for the coefficients (the complete derivation is provided below):

$$c_1 = \frac{\exp(t-s)}{2} \qquad t < s$$

$$\phantom{c_1} = \frac{\exp(s-t)}{2} \qquad t > s \qquad (42)$$

$$c_2 = \frac{k}{\lambda}\left[\frac{\exp[(t-s)(1-\lambda)]}{2-\lambda} - \frac{\exp(t-s)}{2}\right] \qquad t < s$$

$$\phantom{c_2} = \frac{k}{2}\frac{\exp(s-t)}{2-\lambda} \qquad t > s \qquad (43)$$

$$c_3 = \frac{k}{2}\frac{\exp(t-s)}{2-\lambda} \qquad t < s$$

$$\phantom{c_3} = \frac{k}{\lambda}\left[\frac{\exp[(s-t)(1-\lambda)]}{2-\lambda} - \frac{\exp(s-t)}{2}\right] \qquad t > s \qquad (44)$$

$$c_4 = \frac{k^2}{\lambda}\left[\frac{\exp[(1-\lambda)(t-s)]}{2(1-\lambda)(2-\lambda)} - \frac{\exp(t-s)}{2(2-\lambda)}\right] \qquad t < s$$

$$\phantom{c_4} = \frac{k^2}{\lambda}\left[\frac{\exp[(1-\lambda)(s-t)]}{2(1-\lambda)(2-\lambda)} - \frac{\exp(s-t)}{2(2-\lambda)}\right] \qquad t > s. \qquad (45)$$

 

In this work, we focus on the equal-time covariance, $\Sigma$. Evaluating those coefficients at $t = s$ yields the final expression in Eq 10. The equal-time covariance could have been equivalently derived by solving the Lyapunov equation associated with Eq 1, which reads:

$$(k\, \boldsymbol{m}\boldsymbol{n}^\top - I)\Sigma + \Sigma(k\, \boldsymbol{n}\boldsymbol{m}^\top - I) + \bar{\Sigma}_{\text{inp}} = 0. \tag{46}$$

It is easy to verify that the expression for the equal-time covariance provided in Eq 10 satisfies the Lyapunov equation above.

**Coefficient $c_1$** For $t < s$:

$$c_1 = \int_0^t \exp[-(t + s - 2u)]\, \mathrm{d}u = \frac{\exp(t - s) - \exp[-(t + s)]}{2} \rightarrow \frac{\exp(t - s)}{2} \tag{47}$$

as the system approaches the stationary state $(t, s \rightarrow \infty)$. For $t > s$, similarly:

$$c_1 = \int_0^s \exp[-(t + s - 2v)]\, \mathrm{d}v \rightarrow \frac{\exp(s - t)}{2}. \tag{48}$$

**Coefficient $c_2$** For $t < s$:

$$\begin{aligned}
c_2 &= \frac{k}{\lambda} \int_0^t \exp[-(t + s - 2u)] \left\{\exp[\lambda(s - u)] - 1\right\} \mathrm{d}u \\
&= \frac{k}{\lambda} \exp[-(t + s)] \left\{\exp(\lambda s) \int_0^t \exp[u(2 - \lambda)]\, \mathrm{d}u - \int_0^t \exp(2u)\, \mathrm{d}u\right\} \\
&= \frac{k}{\lambda} \exp[-(t + s)] \left\{\frac{\exp[2t + \lambda(s - t)] - \exp(\lambda s)}{2 - \lambda} - \frac{\exp(2t) - 1}{2}\right\} \\
&= \frac{k}{\lambda} \left\{\frac{\exp[(t - s)(1 - \lambda)] - \exp[-t - s(1 - \lambda)]}{2 - \lambda} - \frac{\exp(t - s) - \exp[-(t + s)]}{2}\right\} \\
&\rightarrow \frac{k}{\lambda} \left\{\frac{\exp[(t - s)(1 - \lambda)]}{2 - \lambda} - \frac{\exp(t - s)}{2}\right\}.
\end{aligned} \tag{49}$$

For $t > s$, instead:

$$\begin{aligned}
c_2 &= \frac{k}{\lambda} \int_0^s \exp[-(t + s - 2v)] \left\{\exp[\lambda(s - v)] - 1\right\} \mathrm{d}v \\
&= \frac{k}{\lambda} \exp[-(t + s)] \left\{\exp(\lambda s) \int_0^s \exp[v(2 - \lambda)]\, \mathrm{d}v - \int_0^s \exp(2v)\, \mathrm{d}v\right\} \\
&= \frac{k}{\lambda} \exp[-(t + s)] \left\{\frac{\exp(2s) - \exp(\lambda s)}{2 - \lambda} - \frac{\exp(2s) - 1}{2}\right\} \\
&= \frac{k}{\lambda} \left\{\frac{\exp(s - t) - \exp[-t - s(1 - \lambda)]}{2 - \lambda} - \frac{\exp(s - t) - \exp[-(t + s)]}{2}\right\} \\
&\rightarrow \frac{k}{\lambda} \left\{\frac{\exp(s - t)}{2 - \lambda} - \frac{\exp(s - t)}{2}\right\} = \frac{k}{2} \frac{\exp(s - t)}{2 - \lambda}.
\end{aligned} \tag{50}$$

**Coefficient $c_3$** It follows from the same math as above that, for $t < s$:

$$c_3 = \frac{k}{\lambda} \int_0^t \exp[-(t + s - 2u)] \left\{\exp[\lambda(t - u)] - 1\right\} \mathrm{d}u \rightarrow \frac{k}{2} \frac{\exp(t - s)}{2 - \lambda}, \tag{51}$$

while for $t > s$:

$$c_3 = \frac{k}{\lambda} \int_0^s \exp[-(t+s-2v)] \left\{ \exp[\lambda(s-v)] - 1 \right\} dv \to \frac{k}{\lambda} \left\{ \frac{\exp[(s-t)(1-\lambda)]}{2-\lambda} - \frac{\exp(s-t)}{2} \right\}. \tag{52}$$

**Coefficient $c_4$** For $t < s$:

$$
\begin{aligned}
c_4 &= \frac{k^2}{\lambda^2} \int_0^t \exp[-(t+s-2u)] \left\{ \exp[\lambda(t-u)] - 1 \right\} \left\{ \exp[\lambda(s-u)] - 1 \right\} du \\
&= \frac{k^2}{\lambda^2} \exp[-(t+s)] \left\{ \exp[\lambda(t+s)] \int_0^t \exp[2u(1-\lambda)] du - [\exp(\lambda t) + \exp(\lambda s)] \int_0^t \exp[u(2-\lambda)] du \right. \\
&\quad \left. + \int_0^t \exp(2u) du \right\} \\
&= \frac{k^2}{\lambda^2} \exp[-(t+s)] \left\{ \frac{\exp[t(2-\lambda) + \lambda s] - \exp[\lambda(t+s)]}{2(1-\lambda)} - \frac{\exp(2t) + \exp[2t + \lambda(s-t)] - \exp(\lambda t) - \exp(\lambda s)}{2-\lambda} \right. \\
&\quad \left. + \frac{1}{2}[\exp(2t) - 1] \right\} \\
&\to \frac{k^2}{\lambda^2} \left\{ \frac{\exp[(1-\lambda)(t-s)]}{2(1-\lambda)} - \frac{\exp(t-s) + \exp[(t-s)(1-\lambda)]}{2-\lambda} + \frac{\exp(t-s)}{2} \right\} \\
&= \frac{k^2}{\lambda} \left\{ \frac{\exp[(1-\lambda)(t-s)]}{2(1-\lambda)(2-\lambda)} - \frac{\exp(t-s)}{2(2-\lambda)} \right\},
\end{aligned}
$$

while, using similar algebra, for $t > s$

$$
\begin{aligned}
c_4 &= \frac{k^2}{\lambda^2} \int_0^s \exp[-(t+s-2v)] \left\{ \exp[\lambda(t-v)] - 1 \right\} \left\{ \exp[\lambda(s-v)] - 1 \right\} dv \\
&\to \frac{k^2}{\lambda} \left\{ \frac{\exp[(1-\lambda)(s-t)]}{2(1-\lambda)(2-\lambda)} - \frac{\exp(s-t)}{2(2-\lambda)} \right\}.
\end{aligned} \tag{53}
$$

## 3. One-dimensional inputs

**Scaling analysis** What is the scaling of the different terms in the covariance (Eq 13) with respect to the network size $N$? A quick scaling analysis of Eq 13 yields:

$$
\begin{aligned}
O(\Sigma_{ij}) &= O(u_i)^2 + O(\rho_{nu}) \frac{O(u_i)}{\sqrt{N}} + O(\rho_{nu}) \frac{O(u_i)}{\sqrt{N}} + O(\rho_{nu})^2 \frac{1}{N} \\
&= \frac{1}{N} \left[ 1 + O(\rho_{nu}) + O(\rho_{nu}) + O(\rho_{nu})^2 \right].
\end{aligned} \tag{54}
$$

This shows that the four covariance components display identical scaling with $N$, provided that $O(\rho_{nu}) = O(1)$. This condition holds when the normalized connectivity vector $\boldsymbol{n}$ and input vector $\boldsymbol{u}$ retain a finite overlap for all values of $N$. If the entries of $\boldsymbol{n}$ and $\boldsymbol{u}$ are drawn randomly, this condition is satisfied when the entries are correlated across the two vectors.

**Covariance eigenvalues and eigenvectors** We compute the eigenvalues of the covariance matrix in Eq 13, which has low rank. From [37], we have that the non-zero eigenvalues of a $N \times N$ low-rank matrix expressed in terms of $R$ couples of $\boldsymbol{m}^r$ and $\boldsymbol{n}^r$ vectors as

$$M_{ij} = \sum_{r=1}^{R} m_i^r n_j^r \tag{55}$$

are identical to the eigenvalues of the $R \times R$ reduced matrix defined by

$$M_{rs}^{\text{red}} = \boldsymbol{m}^{r\top}\boldsymbol{n}^s. \tag{56}$$

We apply this rule to Eq 13, yielding the reduced matrix:

$$\frac{1}{2}\begin{pmatrix} 1 & \rho_{mu} & 1 & \rho_{mu} \\ \alpha\rho_{nu} & \alpha\rho_{nu}\rho_{mu} & \alpha\rho_{nu} & \alpha\rho_{nu}\rho_{mu} \\ \alpha\rho_{nu}\rho_{mu} & \alpha\rho_{nu} & \alpha\rho_{nu}\rho_{mu} & \alpha\rho_{nu} \\ \beta\rho_{nu}^2\rho_{mu} & \beta\rho_{nu}^2 & \beta\rho_{nu}^2\rho_{mu} & \beta\rho_{nu}^2 \end{pmatrix} \tag{57}$$

with $\alpha = k/(2-\lambda)$ and $\beta = k^2/[(2-\lambda)(1-\lambda)]$. This matrix has clearly only two non-zero eigenvalues. We therefore use the reduction scheme once again, yielding the further reduced $2 \times 2$ matrix:

$$\frac{1}{2}\begin{pmatrix} 1 + \alpha\rho_{nu}\rho_{mu} & \rho_{mu} + \alpha\rho_{nu} \\ \alpha\rho_{nu} + \beta\rho_{nu}^2\rho_{mu} & \alpha\rho_{nu}\rho_{mu} + \beta\rho_{nu}^2 \end{pmatrix}. \tag{58}$$

From this matrix, eigenvalues can easily be computed, resulting in Eq 14.

We quantify dimensionality using the participation ratio:

$$\mathcal{D} = \frac{\left(\sum_{i=1}^N \mu_i\right)^2}{\sum_{i=1}^N \mu_i^2}. \tag{59}$$

Using Eq 14,

$$\mathcal{D} = \left[1 - 2\frac{\rho_{nu}^2(1-\rho_{mu}^2)(\beta-\alpha^2)}{(1 + 2\alpha\rho_{mu}\rho_{nu} + \beta\rho_{nu}^2)^2}\right]^{-1}, \tag{60}$$

which is bounded between 1 and 2.

We now compute the eigenvectors associated with those eigenvalues. We formulate the ansatz:

$$\boldsymbol{v}_\pm = \gamma_\pm \boldsymbol{m} + \boldsymbol{u}. \tag{61}$$

Note that those vectors are not normalized, but can be normalized by dividing by $(\gamma_\pm^2 + 2\rho_{mu}\gamma_\pm + 1)^{1/2}$. We then impose:

$$\Sigma\boldsymbol{v}_\pm = \mu_\pm \boldsymbol{v}_\pm. \tag{62}$$

With a little algebra, we obtain

$$\Sigma\boldsymbol{v}_\pm = \frac{1}{2}\left[\left(\gamma_\pm\alpha\rho_{nu}\rho_{mu} + \gamma_\pm\beta\rho_{nu}^2 + \alpha\rho_{nu} + \beta\rho_{nu}^2\rho_{mu}\right)\boldsymbol{m} + \left(\gamma_\pm\rho_{mu} + \gamma_\pm\alpha\rho_{nu} + 1 + \alpha\rho_{nu}\rho_{mu}\right)\boldsymbol{u}\right]. \tag{63}$$

Combining this with Eq 62 yields a system of two equations for $\gamma_\pm$ and $\mu_\pm$:

$$\gamma_\pm\mu_\pm = \frac{1}{2}\left(\gamma_\pm\alpha\rho_{nu}\rho_{mu} + \gamma_\pm\beta\rho_{nu}^2 + \alpha\rho_{nu} + \beta\rho_{nu}^2\rho_{mu}\right)$$
$$\mu_\pm = \frac{1}{2}\left(\gamma_\pm\rho_{mu} + \gamma_\pm\alpha\rho_{nu} + 1 + \alpha\rho_{nu}\rho_{mu}\right). \tag{64}$$

Using the second equation together with Eq 14, we find:

$$\gamma_{\pm} = \frac{2\mu_{\pm} - 1 - \alpha\rho_{nu}\rho_{mu}}{\rho_{mu} + \alpha\rho_{nu}}$$

$$= \frac{1}{2(\rho_{mu} + \alpha\rho_{nu})}\left[-1 + \beta\rho_{nu}^2 \pm \sqrt{\left(1 + 2\alpha\rho_{nu}\rho_{mu} + \beta\rho_{nu}^2\right)^2 - 4\rho_{nu}^2(1 - \rho_{mu}^2)(\beta - \alpha^2)}\right].$$

(65)

Note that this equation is ill-defined for $\gamma_+$ when $\rho_{mu} = \rho_{nu} = 0$, for which $\alpha = \beta = 0$. In that case, one can use the first equation in Eq 64 to see that $\gamma_{\pm} = 0$.

**Simulations detail** In Fig 2, we simulated RNNs of size $N = 100$. To construct external inputs and recurrent connectivity, we first generated a set of three random orthonormal vectors $\{z^i\}_{i=1,2,3}$. We then set: $m = z^1$, $n = z^2$, implying that $\rho_{mn} = 0$. In Fig 2C, 2D and 2E, we then set $u = z^3$, implying that $\rho_{nu} = \rho_{mu} = 0$. In Fig 2F, 2G and 2H, we set instead $u = n$, implying that $\rho_{nu} = 1$ and $\rho_{mu} = 0$. In all figures, we fix $k = 2$.

## 4. High-dimensional inputs

**Scaling analysis** The scaling with respect to the network size $N$ of Eq 17 is given by:

$$O(\Sigma_{ij}) = \delta_{ij} + \frac{1}{N} + \frac{1}{N} + \frac{1}{N}.$$

(66)

The equation above shows that, in the case of high-dimensional stochastic inputs, the diagonal entries of the covariance matrix, which quantify single-neuron variances, are significantly larger in amplitude than the off-diagonal entries, which quantify cross-covariances among neurons. This is due to the first term of the covariance (Eq 17), that is generated from local activity (Eq 7), being diagonal and having much larger amplitude than the other three. Eq 66 also indicates that the remaining terms, which express different combinations of the $m$ and $n$ vectors, are characterized by a similar scaling.

**Covariance eigenvalues and eigenvectors** We compute the non-zero eigenvalues of the low-rank component of the covariance matrix, $\Sigma^{lr}$ (Eq 22), following again [37], and arriving at the reduced matrix:

$$\frac{k}{2-\lambda}\begin{pmatrix} \rho_{mn} & 1 & \rho_{mn} \\ 1 & \rho_{mn} & 1 \\ k/(1-\lambda) & \rho_{mn}k/(1-\lambda) & k/(1-\lambda) \end{pmatrix}.$$

(67)

The last two rows are linearly dependent, implying that the original matrix is rank-two. By reducing the matrix further, we get to:

$$\frac{k}{2-\lambda}\begin{pmatrix} \rho_{mn} & 1 \\ 1 + \rho_{mn}k/(1-\lambda) & \rho_{mn} + k/(1-\lambda) \end{pmatrix}$$

(68)

whose eigenvalues are given by Eq 23.

To compute the eigenvectors associated with those eigenvalues, we formulate the ansatz:

$$v_{\pm} = \gamma_{\pm}m + n,$$

(69)

where the normalization factor is in this case given by $(\gamma_{\pm}^2 + 2\rho_{mn}\gamma_{\pm} + 1)^{1/2}$. We then impose:

$$\Sigma^{lr}v_{\pm} = \mu_{\pm}^{lr}v_{\pm}.$$

(70)

With a little algebra, we obtain

$$\Sigma \boldsymbol{v}_\pm = \frac{k}{2-\lambda} \left[ \left( \gamma_\pm \rho_{mn} + 1 + \frac{k}{1-\lambda}\gamma_\pm + \frac{k}{1-\lambda}\rho_{mn} \right) \boldsymbol{m} + (\gamma_\pm + \rho_{mn}) \boldsymbol{n} \right]. \tag{71}$$

Combining it with Eq 70, we see that

$$\begin{aligned}
\gamma_\pm &= \frac{2-\lambda}{k}\mu_\pm^{\text{lr}} - \rho_{mn} \\
&= \frac{1}{2} \left[ \frac{k}{1-\lambda} \pm \sqrt{ \left( 2\rho_{mn} + \frac{k}{1-\lambda} \right)^2 + 4(1 - \rho_{mn}^2)} \right].
\end{aligned} \tag{72}$$

**Limiting cases** We start considering the case of maximally symmetric recurrent connectivity. To derive Eq 25, we use $\rho_{mn} \to 1/k$ and $\lambda \to 1$, and then only keep the diverging terms. Similarly, one gets

$$\gamma_\pm \simeq \frac{1}{2} \left[ \frac{k}{1-\lambda} \pm \left| \frac{k}{1-\lambda} \right| \right], \tag{73}$$

implying that $\gamma_+$ diverges as $\lambda \to 1$.

We then consider the opposite scenario: using $\rho_{mn} \to -1$ and $\lambda \to -k$ in Eq 23, we have

$$\mu_\pm^{\text{lr}} = \frac{k}{2(2+k)} \left[ -2 + \frac{k}{1+k} \pm \left| -2 + \frac{k}{1+k} \right| \right]. \tag{74}$$

The argument of the absolute value on the right-hand side is negative. Therefore we have: $\mu_+^{\text{lr}} = 0$, implying that $\mu_+^{\text{lr}}$ is again equal to $\mu_{\text{ref}}$. Instead:

$$\mu_-^{\text{lr}} = \frac{k}{2+k} \left[ -2 + \frac{k}{1+k} \right] = -\frac{k}{1+k} \tag{75}$$

which is a negative number smaller than $-1$. When $k \to \infty$ (strong recurrent connectivity), $\mu_- \to -1$. Correspondingly,

$$\mu_- = \frac{1}{2}\frac{1}{1+k}, \tag{76}$$

which is a positive number that is always smaller than the reference value, and converges to zero for very large $k$. We also have:

$$\gamma_\pm = \frac{1}{2} \left[ \frac{k}{1+k} \pm \left| -2 + \frac{k}{1+k} \right| \right] \tag{77}$$

from which $\gamma_+ = 1$, and

$$\gamma_- = \frac{1}{2} \left[ \frac{2k}{1+k} - 2 \right] = -\frac{1}{1+k}. \tag{78}$$

Note than the latter is negative, and vanishes for $k \to \infty$. This implies that the eigenvector $\boldsymbol{v}_-$ has a small and negative component along connectivity vector $\boldsymbol{m}$, and a larger and positive component along connectivity vector $\boldsymbol{n}$.

**Simulations detail** In Fig 4, we simulated RNNs of size $N = 50$. To construct external inputs and recurrent connectivity, we first generated a set of two orthonormal vectors $\{z^i\}_{i=1,2}$. We then set: $m = z^1$, and

$$n = \rho_{mn} z^1 + \sqrt{1 - \rho_{mn}^2} z^2. \tag{79}$$

In Fig 4A, 4B and 4C, we then set $\rho_{mn} = -0.5$. In Fig 4D, 4E and 4F, we then set $\rho_{mn} = 0.3$. For simplicity, we also set $U = I$.

## 5. Origin of local and recurrent activity terms

To clarify the origin and implications of the separation of activity in two terms (Eq 7), we consider the discrete-time approximation of Eq 1:

$$x(t + \tau) = (1 - \tau) x(t) + \tau W x(t) + \tau^{1/2} \xi(t + \tau) \tag{80}$$

which is accurate in the $\tau \to 0$ limit. We have used the short-hand notation $\xi(t) = U\chi(t)$.

Starting from a simple initial condition $x(0) = 0$, we have $x(\tau) = \tau^{1/2} \xi(\tau)$, and therefore

$$\begin{aligned} x(2\tau) &= (1 - \tau) x(\tau) + \tau W x(\tau) + \tau^{1/2} \xi(2\tau) \\ &= (1 - \tau)\tau^{1/2} \xi(\tau) + \tau^{3/2} W\xi(\tau) + \tau^{1/2} \xi(2\tau). \end{aligned} \tag{81}$$

In analogy with Eq 7, we can rewrite this as $x(2\tau) = x^{\text{loc}}(2\tau) + x^{\text{rec}}(2\tau)$ by defining

$$\begin{aligned} x^{\text{loc}}(2\tau) &= (1 - \tau)\tau^{1/2} \xi(\tau) + \tau^{1/2} \xi(2\tau) \\ x^{\text{rec}}(2\tau) &= \tau^{3/2} W\xi(\tau). \end{aligned} \tag{82}$$

Taking one extra step, one can easily show that:

$$\begin{aligned} x^{\text{loc}}(3\tau) &= (1 - \tau)^2\tau^{1/2} \xi(\tau) + (1 - \tau)\tau^{1/2} \xi(2\tau) + \tau^{1/2} \xi(3\tau) \\ x^{\text{rec}}(3\tau) &= \tau^{5/2} W^2\xi(\tau) + \tau^{3/2} W\xi(2\tau) + (1 - \tau)\tau^{3/2} W\xi(\tau). \end{aligned} \tag{83}$$

We highlight the following points. First, both activity terms $x^{\text{loc}}$ and $x^{\text{rec}}$ are expressed as sums over terms that depend on the stochastic input at different time points. Unlike $x^{\text{loc}}$, $x^{\text{rec}}$ depends only on inputs from previous time steps (after passing through the recurrent connectivity). Second, the leak in the dynamics (Eq 1), which contributes to all the terms proportional to $(1 - \tau)$, feeds into both activity terms, $x^{\text{loc}}$ and $x^{\text{rec}}$. Third, removing the leak from the dynamics modifies the temporal properties of activity, but does not affect its two-contribution structure (Eq 7). This can be easily verified by noting that removing the leak is equivalent to transforming all $(1 - \tau)$ terms into 1 in the equations above.

Finally, we consider one specific type of discrete-time dynamics, obtained by setting $\tau = 1$ (very large updates, corresponding to very fast leak and dynamics). This type of model is of particular interest because it has been widely studied in the context of RNNs with random connectivity [38] and is broadly used in machine learning [39] and statistics [28] applications. For these models, one gets:

$$x^{\text{loc}}(n\tau) = \tau^{1/2}\xi(n\tau), \tag{84}$$

implying that the local activity component loses memory of inputs from previous time steps. However, this is not the case for the recurrent component, for which

$$x^{\text{rec}}(n\tau) = \sum_{m=1}^{n-1} \tau^{(n-m)+\frac{1}{2}} W^{n-m} \boldsymbol{\xi}(m\tau). \tag{85}$$

We conclude that, in discrete-time models with $\tau = 1$, the network's memory of previous inputs is maintained solely through recurrent interactions. The two activity contributions, $x^{\text{loc}}$ and $x^{\text{rec}}$, reflect stochastic inputs at different time points, and are therefore uncorrelated. We therefore expect the activity covariance for these models to take a simpler form than Eq 10, where two of the four terms arise from the cross-covariance between $x^{\text{loc}}$ and $x^{\text{rec}}$.

## 6. Propagator of higher-rank matrices

For a connectivity matrix of rank greater than one (Eq 26), we have:

$$\exp[(W - I)t] = \exp\left[\left(k \sum_{r=1}^{R} \boldsymbol{m}^r \boldsymbol{n}^{r\top}\right) t\right] \exp(-t). \tag{86}$$

Evaluating this matrix exponential is generally challenging for a generic low-rank matrix $W$. However, we can consider a specific subclass of these matrices, where the connectivity vectors $\boldsymbol{m}^r$ and $\boldsymbol{n}^r$ associated with different rank-one components are mutually orthogonal:

$$\boldsymbol{n}^{r\top} \boldsymbol{m}^{r'} = 0 \tag{87}$$

for $r \neq r'$. For these matrices, the nonzero eigenvalues are given by $\lambda^r = k\rho_{m^r n^r}$ for $r = 1, \dots, R$, with corresponding eigenvectors $\boldsymbol{m}^r$. This class includes, as a special case, the low-rank matrices studied in [12] and [49], for which all eigenvalues vanish. It does not include low-rank matrices with complex eigenvalues.

Eq 87 implies that all rank-one components commute, leading to:

$$
\begin{aligned}
\exp\left[\left(k \sum_{r=1}^{R} \boldsymbol{m}^r \boldsymbol{n}^{r\top}\right) t\right] &= \prod_{r=1}^{R} \exp\left(k \, \boldsymbol{m}^r \boldsymbol{n}^{r\top} t\right) \\
&= \prod_{r=1}^{R} \left[ I + \frac{\exp(\lambda^r t) - 1}{\lambda^r} k \, \boldsymbol{m}^r \boldsymbol{n}^{r\top} \right] \\
&= I + \sum_{r=1}^{R} \frac{\exp(\lambda^r t) - 1}{\lambda^r} k \, \boldsymbol{m}^r \boldsymbol{n}^{r\top}.
\end{aligned}
\tag{88}
$$

Thus, the propagator takes the form:

$$\exp[(W - I)t] = \exp\left[ I + \sum_{r=1}^{R} \frac{\exp(\lambda^r t) - 1}{\lambda^r} k \, \boldsymbol{m}^r \boldsymbol{n}^{r\top} \right] \exp(-t). \tag{89}$$

This expression assumes that $\lambda^r \neq 0$ for all $r$. If the eigenvalue associated with a given rank-one component vanishes, the corresponding term in the sum in Eq 89 must be replaced with the form given in Eq 36.

## 7. Covariance of higher-rank networks

To compute activity covariance, we start from Eq 38, use again Eq 89, and we arrive at

$$\Sigma(t,s) = c_1(t,s)\bar{\Sigma}_{\text{inp}} + \sum_{r=1}^{R}\left[c_2^r(t,s)\,\bar{\Sigma}_{\text{inp}}\boldsymbol{n}^r\boldsymbol{m}^{r\top} + c_3^r(t,s)\,\boldsymbol{m}^r\boldsymbol{n}^{r\top}\bar{\Sigma}_{\text{inp}}\right] + \sum_{r=1}^{R}\sum_{r'=1}^{R}c_4^{rr'}(t,s)\boldsymbol{m}^r\boldsymbol{n}^{r\top}\bar{\Sigma}_{\text{inp}}\boldsymbol{n}^{r'}\boldsymbol{m}^{r'\top}.$$

(90)

Using algebra similar to that in Methods 2, one can show that $c_1(t,s)$ is given by Eq 42, while $c_2^r(t,s)$ and $c_3^r(t,s)$ obey Eqs 43 and 44 (with $\lambda$ replaced by $\lambda^r$). Finally, $c_4^{rr'}(t,s)$ is given by:

$$
\begin{aligned}
c_4^{rr'} &= \frac{k^2}{\lambda^{r'}}\left[\frac{\exp[(1-\lambda^{r'})(t-s)]}{(2-\lambda^r-\lambda^{r'})(2-\lambda^{r'})} - \frac{\exp(t-s)}{2(2-\lambda^r)}\right] \quad t<s \\
&= \frac{k^2}{\lambda^r}\left[\frac{\exp[(1-\lambda^r)(s-t)]}{(2-\lambda^r-\lambda^{r'})(2-\lambda^{r'})} - \frac{\exp(s-t)}{2(2-\lambda^{r'})}\right] \quad t>s
\end{aligned}
$$

(91)

which becomes equal to Eq 45 for $\lambda^r = \lambda^{r'}$. Evaluating the covariance at $t = s$ yields:

$$
\begin{aligned}
\Sigma = \frac{1}{2}\Bigg\{&\bar{\Sigma}_{\text{inp}} + \sum_{r=1}^{R}\frac{k}{2-\lambda^r}\left[\bar{\Sigma}_{\text{inp}}\boldsymbol{n}^r\boldsymbol{m}^{r\top} + \boldsymbol{m}^r\boldsymbol{n}^{r\top}\bar{\Sigma}_{\text{inp}}\right] \\
&+ \sum_{r=1}^{R}\sum_{r'=1}^{R}\frac{k^2(4-\lambda^r-\lambda^{r'})}{(2-\lambda^r-\lambda^{r'})(2-\lambda^r)(2-\lambda^{r'})}\boldsymbol{m}^r\boldsymbol{n}^{r\top}\bar{\Sigma}_{\text{inp}}\,\boldsymbol{n}^{r'}\boldsymbol{m}^{r'\top}\Bigg\}
\end{aligned}
$$

(92)

from which Eq 28 can be derived.

We focus on the case of high-dimensional stochastic inputs (Eq 30). The eigenvalues of the covariance matrix can be computed following Eq 21, with $\mu_i^{\text{lr}}$ being the eigenvalues of the low-rank component

$$\Sigma^{\text{lr}} = \sum_{r=1}^{R}\frac{k}{2-\lambda^r}\left[\boldsymbol{n}^r\boldsymbol{m}^{r\top} + \boldsymbol{m}^r\boldsymbol{n}^{r\top}\right] + \sum_{r=1}^{R}\sum_{r'=1}^{R}\frac{\rho_{n^r n^{r'}}\,k^2(4-\lambda^r-\lambda^{r'})}{(2-\lambda^r-\lambda^{r'})(2-\lambda^r)(2-\lambda^{r'})}\boldsymbol{m}^r\boldsymbol{m}^{r'\top}.$$

(93)

To compute the eigenvalues of this matrix, we first observe that we can re-write it as

$$\Sigma^{\text{lr}} = \sum_{r=1}^{R}\left[\boldsymbol{n}^r(\alpha^r\boldsymbol{m}^r)^\top + \boldsymbol{m}^r\left(\alpha^r\boldsymbol{n}^{r\top} + \sum_{r'=1}^{R}\rho_{n^r n^{r'}}\beta^{rr'}\boldsymbol{m}^{r'\top}\right)\right]$$

(94)

where we have defined

$$\alpha^r = \frac{k}{2-\lambda^r}$$

(95)

$$\beta^{rs} = \frac{k^2(4-\lambda^r-\lambda^s)}{(2-\lambda^r-\lambda^s)(2-\lambda^r)(2-\lambda^s)}.$$

(96)

From this formulation, we can derive a $2R \times 2R$ reduced matrix [37], constructed by concatenating $2 \times 2$ blocks horizontally and vertically. These blocks take the values

$$B^{rs} = \begin{pmatrix} \delta^{rs}\rho_{m^r n^r}\alpha^r & \rho_{n^r n^s}\alpha^s + \rho_{m^r n^r}\rho_{n^r n^s}\beta^{sr} \\ \rho_{m^r m^s}\alpha^s & \delta^{rs}\rho_{m^r n^r}\alpha^r + \sum_{s'}\rho_{m^r m^{s'}}\rho_{n^s n^{s'}}\beta^{ss'} \end{pmatrix}$$

(97)

where $\delta^{rs}$ denotes the Kronecker delta. Computing the eigenvalues of this reduced matrix yields the eigenvalues of $\Sigma^{\mathrm{lr}}$.

These eigenvalues take particularly simple values for a subclass of low-rank matrices in which $\rho_{m^r n^r}$ can take arbitrary values, ranging between $-1$ and $1/k$, while $\rho_{m^r m^s}$ and $\rho_{n^r n^s}$ vanish for all $r \neq s$ (and are equal to 1 otherwise). This case corresponds to different rank-one components operating in orthogonal subspaces. For these matrices, all entries of off-diagonal blocks $B^{rs}$ for $r \neq s$ vanish, while the diagonal blocks take the form:

$$B^{rr} = \begin{pmatrix} \rho_{m^r n^r} \alpha^r & \alpha^s + \rho_{m^r n^r} \beta^{rr} \\ \alpha^r & \rho_{m^r n^r} \alpha^r + \beta^{rr} \end{pmatrix} \tag{98}$$

for $r = 1, 2$. Thus, the eigenvalues of the total reduced matrix coincide with those of the $2 \times 2$ diagonal blocks. Using the identity

$$\beta^{rr} = \frac{k^2}{(1 - \lambda^r)(2 - \lambda^r)}, \tag{99}$$

one can show that each diagonal block is associated with a pair of eigenvalues $\mu_{\pm}^r$, obeying Eq 23, where $\lambda$ is replaced by $\lambda^r$ and $\rho_{mn}$ by $\rho_{m^r n^r}$. The eigenvectors corresponding to $\mu_{\pm}^r$ are linear combinations of the connectivity vectors $\boldsymbol{m}^r$ and $\boldsymbol{n}^r$, similar to Eq 24.

In Fig 5, we consider rank-2 connectivity matrices with two specific parametrizations. The first corresponds to the simplified case described above, where $\rho_{m^1 n^1}$ and $\rho_{m^2 n^2}$ take arbitrary values, while $\rho_{m^1 m^2}$ and $\rho_{n^1 n^2}$ are set to zero. The second parametrization, in contrast, involves setting $\rho_{m^1 n^1}$ and $\rho_{m^2 n^2}$ to zero while allowing $\rho_{m^1 m^2}$ and $\rho_{n^1 n^2}$ to take arbitrary values within the range $[-1,1]$. Unlike the first case, this choice leads to a connectivity matrix with vanishing eigenvalues. Moreover, in this regime, the covariance eigenvectors are linear combinations of all four connectivity vectors: $\boldsymbol{m}^1$, $\boldsymbol{n}^1$, $\boldsymbol{m}^2$, and $\boldsymbol{n}^2$.

**Simulations detail** In Fig 5F, we simulated RNNs of size $N = 50$. To construct external inputs and recurrent connectivity, we first generated a set of four normalized orthogonal vectors $\{\boldsymbol{z}^i\}_{i=1,\dots,4}$. We then set: $\boldsymbol{m}^1 = \boldsymbol{z}^1$, $\boldsymbol{m}^2 = \boldsymbol{z}^2$, and

$$\begin{aligned} \boldsymbol{n}^1 &= \rho_{m^1 n^1} \boldsymbol{z}^1 + \sqrt{1 - \rho_{m^1 n^1}^2} \, \boldsymbol{z}^3 \\ \boldsymbol{n}^2 &= \rho_{m^2 n^2} \boldsymbol{z}^2 + \sqrt{1 - \rho_{m^2 n^2}^2} \, \boldsymbol{z}^4 \end{aligned} \tag{100}$$

We fix $\rho_{m^1 n^1} = 0.1$, $\rho_{m^2 n^2} = -0.3$. We also set $U = I$.

## 8. Excitatory-inhibitory network

In the case of one-dimensional inputs, we parametrize the input vector as $\boldsymbol{u} = (u_{\mathrm{E}}, u_{\mathrm{I}}) = (\cos(\theta), \sin(\theta))$, where $\theta$ varies between 0 and $\pi$. Special cases of interest are: $\theta = 0$ (input only to E), $\theta = \pi/4$ (equal inputs to E and I), $\theta = \pi/2$ (input only to I), $\theta = 3\pi/4$ (opposite-sign, otherwise equal input to E and I). This results in the following overlaps:

$$\begin{aligned} \rho_{mn} &= \frac{1 - g}{\sqrt{2(g^2 + 1)}} \\ \rho_{nu} &= \frac{1}{\sqrt{g^2 + 1}} \left[ \cos(\theta) - g \sin(\theta) \right] \\ \rho_{mu} &= \frac{1}{\sqrt{2}} \left[ \cos(\theta) + \sin(\theta) \right]. \end{aligned} \tag{101}$$

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
