## [Decision Letter · Decision Letter 0]

30 Jun 2025

PCOMPBIOL-D-25-00965

Stochastic activity in low-rank recurrent neural networks

PLOS Computational Biology

Dear Dr. Mastrogiuseppe,

Thank you for submitting your manuscript to PLOS Computational Biology. After careful consideration, we are pleased to inform you that we would like to proceed with a minor revision. We invite you to submit a revised version of your manuscript that addresses the points raised by the reviewers.

Please submit your revised manuscript within 30 days Aug 30 2025 11:59PM. If you will need more time than this to complete your revisions, please reply to this message or contact the journal office at ploscompbiol@plos.org. Please include the following items when submitting your revised manuscript:

We look forward to receiving your revised manuscript.

Kind regards,

Oren Shriki, PhD

Academic Editor

PLOS Computational Biology

Marieke van Vugt

Section Editor

PLOS Computational Biology

**Additional Editor Comments :**

We suggest adding a brief discussion in the Discussion section on the relationship between learning and emergent connectivity. In particular, it would be illuminating to consider how the dimensionality of inputs during learning—low-dimensional versus high-dimensional—may influence the rank of the resulting connectivity matrix.

We also note the following minor points for your attention:

In the paragraph following Equation (2), the manuscript states:

“Our analysis focuses on the activity generated by this model in the stationary regime, which is reached provided that the real part of all eigenvalues of W is less than 1.”

It may help readers if you clarify that this condition ensures the real part of the eigenvalues governing the dynamics is negative, thereby stabilizing the system.

The use of in Equation (11) may be confusing, given that is already used for network activity earlier in the manuscript. Consider using a different symbol to avoid ambiguity.

**Journal Requirements:**

At this stage, the following Authors/Authors require contributions: Francesca Mastrogiuseppe, Joana Carmona, and Christian Machens. Please ensure that the full contributions of each author are acknowledged in the "Add/Edit/Remove Authors" section of our submission form.

4) Thank you for indicating that "The authors declare no competing interests." Please state "The authors have declared that no competing interests exist."

**Reviewers' comments:**

Reviewer's Responses to Questions

Reviewer #1: In this work the authors study the dynamics of low-rank linear recurrent neural networks driven by stochastic inputs. The dimensionality of the input is varied and the interplay between input dimensionality and weight structure is studied. This leads to some important effects on the dimensionality of the dynamics themselves, which are shown to be high-dimensional even in cases where the weights are low-rank. This contrasts with previous studies by the authors and others, and they do a good job explaining why the differences occur.

I found the modeling interesting and the paper very well-written. Overall, I think the paper is well-suited for PLOS CB.

One main weakness, in my opinion, is the restriction to just linear dynamics. Can similar qualitative behavior can occur in nonlinear networks? Another weakness is that the connections to biology aren't at the forefront of the paper. For the most part it concerns the mathematical results, although a nice application to very simple E/I networks appears at the end of the results. The emphasis is on computation over biology, so if the biological connections could be made stronger in the discussion or elsewhere, that would help.

Besides the inclusion of a link to the code that ran the experiments, I have otherwise minor suggestions for the authors to improve the paper.

The parallel study by Wan and Rosenbaum that looked at low-rank linear dynamics is mentioned in the discussion. The work here should be compared and contrasted with that in more detail. Please clarify how "their mathematical framework differs" and the "results are qualitatively consistent". A full paragraph in the discussion is likely needed, and Wan and Rosenbaum might be referenced in the introduction.

I'm not sure I see Section 4.1 as being very helpful as part of the main text. Most of the mathematics in here seems more suited to the appendix. There's a lot of notation introduced to describe the filtered signals that doesn't appear again. On the other hand, the last two paragraphs of that section are helpful for interpreting Eqns (7-9); it makes sense to keep those in the main text.

Minor points below

------------------

Abstract: "remains however not fully understood" awkward

page 3, 4th para: sentence "While the dependence..." has em-dashes formatted wrong

page 6, 1st para: This dimensionality measure goes back much further than ref 3. It was originally introduced in papers by Yule (studying linguistics) and later Simpson (studying ecology) in the 1940s and is equivalent to the Renyi 2-entropy.

page 6, 2nd para: "estimated" should be "computed" since there is no approximation here.

page 6, 3rd para: 2nd sentence was confusing to me and would be better phrased like "Within the m-u plane, the eigenvectors v_\pm define the principle components of the activity fluctuations".

pg 7, sec 4.1, 2nd para: Define "second covariance term"

page 8, 1st para: The notation \hat\chi_i(\Lambda) hides the t-dependence. This isn't a transform into a timescale variable but still a time-dependent function. I would keep the t-dependence and and also be explicity that the averages <...> are over the time variable. Earlier, these are used to be averages over input realizations.

page 11, eqn 27: superscript notation \lambda^r looks like a power; in fact this is intermixed in this equation with variables raised to powers. To clarify where you are using superscripts, I suggest writing \lambda^{(r)}, etc.

Reviewer #2: In this paper Mastrogiuseppe et al. extend the low-rank dynamical framework developed initially by the first author in a very logical and valuable way. I found this paper adds a crucial piece to this body of work, while being thorough and interesting. Overall I don’t have much to add, only a few very minor comments and suggestions, mostly on exposition or details.

Section 6: I would move the definition of sum and difference vectors from the fig legend to the text, for clarity. In the figure, perhaps consider putting an indication of E = 0; I = 90 (e.g. on the plot axes), or a little schematic to the left (e.g. a vector diagram or so, but this might be overkill), just to help visualization. In the second-to-last paragraph, maybe include n and m as parentheticals to remind the reader of the relationships where you say “For input vectors strongly aligned with the difference direction (n) … Instead, for inputs strongly aligned with the sum direction (m)…”. In paragraph 3, add a citation(s) in the last sentence beginning “Consistent with previous work…”

I didn’t recall seeing a link to the published code in the paper, so please include one in the methods (apologies if I just missed it).

A few typos I noticed:

Intro, paragraph 5: “…low-rank RNNs in response to this type of INPUT is …”

After eq. 5, definition of propagator operator, should (W - 1) be written (W - I)

Pg. 9, first paragraph: “Along OF all other directions, including v_+ …” - remove “of”

Fig. 5 caption: “… from simulated activity and THE THE theoretically-estimated …” (double “the”)

Section 6, last sentence: “… similarly to THE rank one case …”

Section 6, paragraph 3: “… a natural framework for studying INPUT amplification”

Reviewer #3: The manuscript investigates the role of stochastic, high-dimensional inputs in shaping the dynamics of low-rank networks. This is an important problem to study, since previous papers mainly focused on deterministic low-dimensional inputs to low-rank networks that are unrealistic for modeling brain dynamics. The authors find that high-dimensional inputs could give rise to high-dimensional dynamics depending on the network structure. The manuscript is clearly written, and especially the analytical derivations are thoroughly described and easy to follow. The findings provide new insights into how the input structure shapes the dynamics of low-rank networks and will be valuable to a broad range of neuroscientists interested in how input and network structure jointly shape neural dynamics.

I just have a few questions and minor suggestions outlined below.

Questions:

1. How much do the findings related to high-dimensional inputs depend on the fact that all inputs have the same mean and variance? For example, would dependency on m and n change if specific input dimensions are stronger or more variable?

2. How do the model predictions change as the rank of the connectivity increases? Would, at a certain rank, the role of input in shaping high-dimensional dynamics vanish, and dynamics become dominated by recurrent components rather than input?

3. In a real brain, how could one distinguish the role of high-dimensional inputs versus high-rank connectivity in shaping high-dimensional dynamics? Does the proposed theory provide any testable predictions? If such predictions exist, it would be interesting to mention them in the Discussion.

Minor suggestions:

1. To make the paper accessible to a broader audience, it would be helpful if authors could provide a summary of the findings in the form of a table or graphical abstract. While the analytical derivations are described in detail, a non-expert audience may prefer to get a general understanding of dependencies without going through the details of the calculations.

2. In figures 2, 4, 5, and 6, it would be helpful to add the variable describing the colorbars directly to the figure, instead of only mentioning it in the caption.

3. If Section 3 only discusses one-dimensional inputs, it would be more informative to change the title of this section to “one-dimensional stochastic inputs”.

4. In the last paragraph of the Discussion about how trial-to-trial variability relates to network structure, the authors could also refer to the following experimental and modeling studies relating trial-to-trial variability to the neural connectivity structure:

- Rosenbaum, R., Smith, M. A., Kohn, A., Rubin, J. E., & Doiron, B. (2017). The spatial structure of correlated neuronal variability. Nature neuroscience, 20(1), 107-114.

- Safavi, S., Dwarakanath, A., Kapoor, V., Werner, J., Hatsopoulos, N. G., Logothetis, N. K., & Panagiotaropoulos, T. I. (2018). Nonmonotonic spatial structure of interneuronal correlations in prefrontal microcircuits. Proceedings of the National Academy of Sciences, 115(15), E3539-E3548.

- [already mentioned in the introduction] Huang, C., Ruff, D. A., Pyle, R., Rosenbaum, R., Cohen, M. R., & Doiron, B. (2019). Circuit models of low-dimensional shared variability in cortical networks. Neuron, 101(2), 337-348.

- Shi, Y. L., Steinmetz, N. A., Moore, T., Boahen, K., & Engel, T. A. (2022). Cortical state dynamics and selective attention define the spatial pattern of correlated variability in neocortex. Nature communications, 13(1), 44.

5. For reproducibility purposes, I recommend that authors provide a README for their shared code that describes what each script does and which figures they reproduce.

**Have the authors made all data and (if applicable) computational code underlying the findings in their manuscript fully available?**

Reviewer #1: **No: **Please include link to repo in final version

Reviewer #2: Yes

Reviewer #3: Yes

PLOS authors have the option to publish the peer review history of their article (what does this mean?). If published, this will include your full peer review and any attached files.

Reviewer #1: **Yes: **Kameron Decker Harris

Reviewer #2: No

Reviewer #3: No

**Figure resubmission:**
---

## [Decision Letter · Decision Letter 1]

28 Jul 2025

Dear Mastrogiuseppe,

We are pleased to inform you that your manuscript 'Stochastic activity in low-rank recurrent neural networks' has been provisionally accepted for publication in PLOS Computational Biology.

Best regards,

Oren Shriki, PhD

Academic Editor

PLOS Computational Biology

Marieke van Vugt

Section Editor

PLOS Computational Biology

Reviewer's Responses to Questions

**Comments to the Authors:**

Reviewer #1: Thank you to the authors for incorporating my suggestions. I am happy with the revision and recommend acceptance.

Reviewer #3: I thank the authors for considering my comments and incorporating them in the manuscript. I support its publication.

**Have the authors made all data and (if applicable) computational code underlying the findings in their manuscript fully available?**

Reviewer #1: Yes

Reviewer #3: Yes

PLOS authors have the option to publish the peer review history of their article (what does this mean?). If published, this will include your full peer review and any attached files.

Reviewer #1: **Yes: **Kameron Decker Harris

Reviewer #3: No

---

## [Editor Report · Acceptance letter]

PCOMPBIOL-D-25-00965R1

Stochastic activity in low-rank recurrent neural networks

Dear Dr Mastrogiuseppe,

I am pleased to inform you that your manuscript has been formally accepted for publication in PLOS Computational Biology. Your manuscript is now with our production department and you will be notified of the publication date in due course.

With kind regards,

Anita Estes
